# Diverse populations of local interneurons integrate into the Drosophila adult olfactory circuit

Nan-Fu Liou [1], Shih-Han Lin[1], Ying-Jun Chen[1], Kuo-Ting Tsai[1], Chi-Jen Yang [1], Tzi-Yang Lin[1,4], Ting-Han Wu[1], Hsin-Ju Lin[1], Yuh-Tarng Chen[1], Daryl M. Gohl [2,5], Marion Silies[2,6] & Ya-Hui Chou [1,3]

*Drosophila* olfactory local interneurons (LNs) in the antennal lobe are highly diverse and variable. How and when distinct types of LNs emerge, differentiate, and integrate into the olfactory circuit is unknown. Through systematic developmental analyses, we found that LNs are recruited to the adult olfactory circuit in three groups. Group 1 LNs are residual larval LNs. Group 2 are adult-specific LNs that emerge before cognate sensory and projection neurons establish synaptic specificity, and Group 3 LNs emerge after synaptic specificity is established. Group 1 larval LNs are selectively reintegrated into the adult circuit through pruning and re-extension of processes to distinct regions of the antennal lobe, while others die during metamorphosis. Precise temporal control of this pruning and cell death shapes the global organization of the adult antennal lobe. Our findings provide a road map to understand how LNs develop and contribute to constructing the olfactory circuit.

[1] Institute of Cellular and Organismic Biology, Academia Sinica, Taipei 11529, Taiwan. [2] Department of Neurobiology, Stanford University, Stanford, CA 94305, USA. [3] Neuroscience Program of Academia Sinica, Academia Sinica, Taipei 11529, Taiwan. [4] Present address: Research Institute of Molecular Pathology (IMP), Vienna Biocenter, Campus-Vienna-Biocenter 1, 1030 Vienna, Austria. [5] Present address: University of Minnesota Genomics Center, 1-210 CCRB, 2231 6th Street SE, Minneapolis, MN 55455, USA. [6] Present address: European Neuroscience Institute, University Medical Center Göttingen, Grisebachstr. 5, 37077 Göttingen, Germany. These authors contributed equally: Nan-Fu Liou, Shih-Han Lin, Ying-Jun Chen Correspondence and requests for materials should be addressed to Y.-H.C. (email: yhchou@gate.sinica.edu.tw)

Developmental or postnatal defects in interneurons have long been associated with neurological disorders[1,2], yet the diversity of these cells makes studying their development and functions in normal and pathological conditions extremely challenging[3–7]. In addition, interneurons often exhibit plasticity and variability[4,8], as well as non-uniform developmental programs. In some vertebrates, interneurons are continuously generated from the postnatal stage to adulthood and sequentially integrate into existing mature circuits[9,10]. All of these features add complexity to wiring principles of interneurons. Therefore, detailed characterization of relatively less complicated circuits in model organisms may illuminate the guiding principles that govern the physiological functions and dysfunction of interneurons.

The *Drosophila* olfactory circuit shares similar organizational principles with those of mammals in the first olfactory information processing center (i.e., olfactory bulb in mammals and antennal lobe in fly) but it is numerically simpler[11–14]. Accumulating evidence has demonstrated that *Drosophila* olfactory local interneurons (LNs) exhibit a high degree of morphological and electrophysiological diversity, and animal-to-animal variability[4,15–23]. In addition to the differences between cells, the molecular properties and morphologies of individual LNs may change in response to alterations in environmental stimuli, such as different food sources or $CO_2$-containing stress odor[24–26]. Because these characteristics of interneurons are similar among

many species, *Drosophila* olfactory LNs may serve as a model to reveal common mechanisms of interneuron development and wiring principles. The wiring logic of LNs was recently examined in first instar larvae, when the larval olfactory system develops[27], as well as in three glomeruli of adult antennal lobe (AL)[28]. However, how and when distinct types of LNs emerge, differentiate, and integrate into the adult olfactory circuit remains unknown.

*Drosophila* develops two olfactory systems in its lifecycle, the larval olfactory system and the adult olfactory system. Similar to adult LNs, larval LNs are morphologically diverse[29]. In this study, we consider larval LNs to be those that first elaborate processes in the larval AL. Accordingly, we designate LNs that only elaborate processes in the adult AL after pupal formation as adult-specific LNs, regardless of their birth timing. We aimed to disentangle the diversity and variability of larval and adult LNs by identifying genetic drivers that could be used to label distinct subtypes of LNs. Observing and altering the fates of these labeled cells allowed us to better understand how and when various types of LNs integrate to the adult olfactory circuit to establish diversity, and whether LNs contribute to the construction of the developing circuit. Overall, we found that LNs in the adult circuit are sequentially recruited to the developing AL, including a subset of larval LNs that are pruned and reintegrated. Precisely controlled pruning and degeneration of LN processes contributes to shaping the global organization and characteristic positions of glomeruli

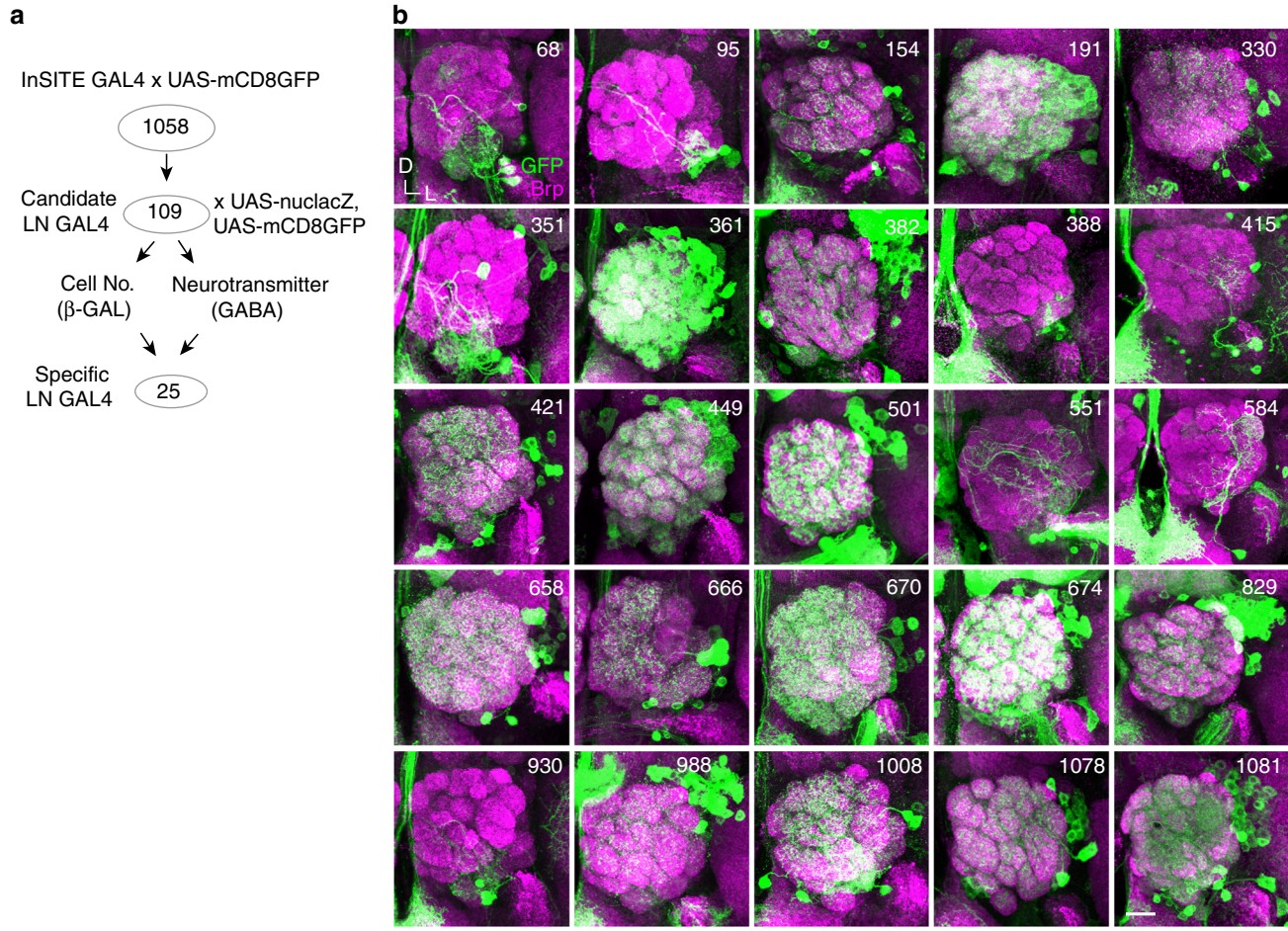

**Fig. 1** Genetic screen for GAL4 drivers that label distinct subtypes of LNs. **a** An overview of the LN GAL4 screen is shown. **b** Projected confocal images show labeled LNs of 25 identified LN GAL4 lines. Adult brains were stained with neuropil markers Bruchpilot (magenta). LNs were visualized by GAL4-driven mCD8GFP (green). The scale bar in this and all figures (unless otherwise indicated) is 20 μm. All ALs are oriented such that the midline is in the left. D: dorsal, L: lateral

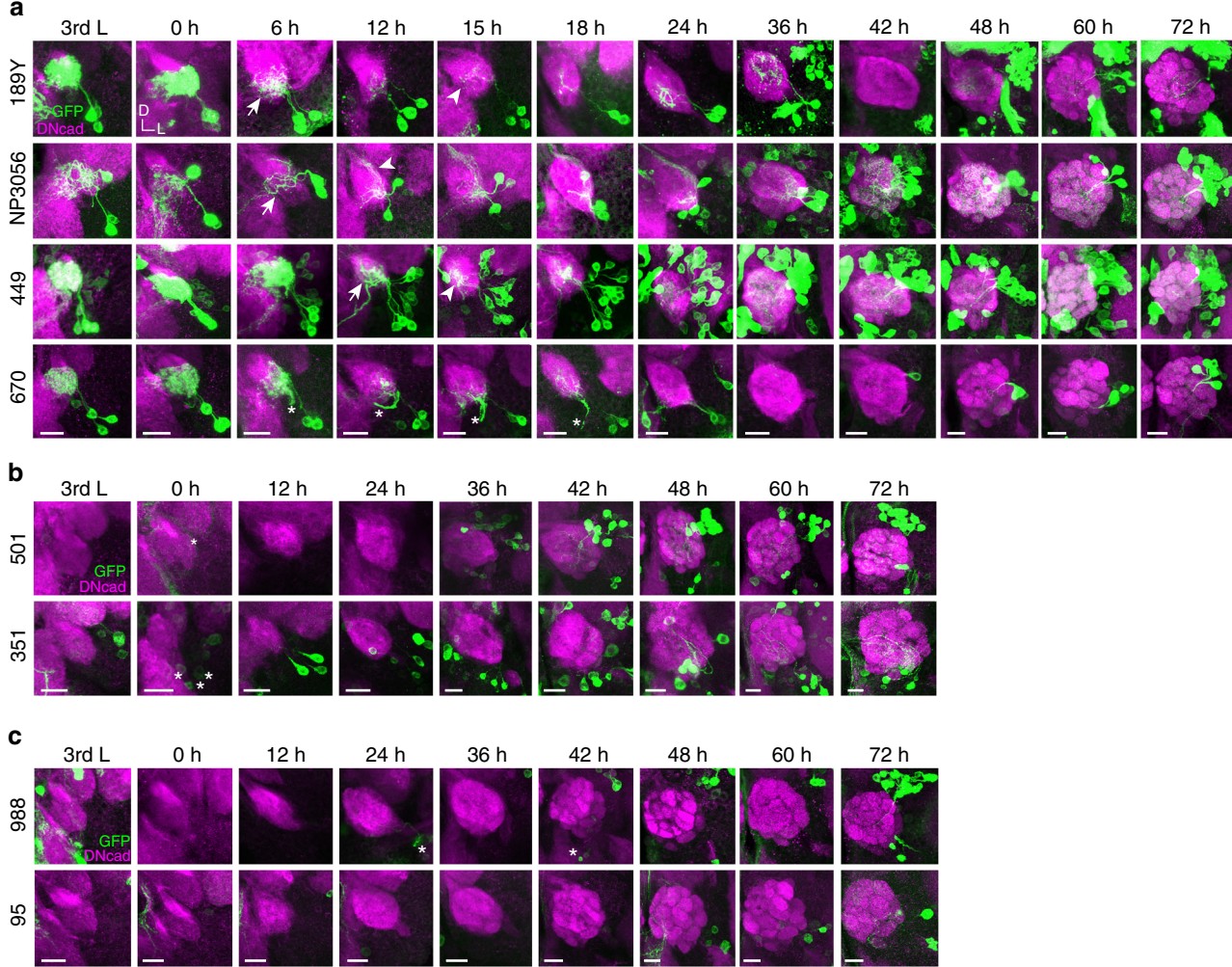

**Fig. 2** The development of LNs during metamorphosis. **a** Group 1 GAL4 drivers that label both larval and adult-specific LNs. **b** Group 2 GAL4 drivers that label adult-specific LNs appearing before 48 h APF. **c** Group 3 GAL4 drivers that label adult-specific LNs appearing after 48 h APF. Projected confocal images show larval and pupal brains stained with neuropil markers DNcad (magenta). LNs were visualized by GAL4-driven mCD8GFP (green). Developmental stages are indicated by 3rd L: third instar larval and hours after puparium formation (APF): pupal stages. Arrows and arrowheads indicate pruning and re-extended processes, respectively. Asterisks denote glia, neurons that are not LNs or their processes. All ALs are oriented such that the midline is in the left. D: dorsal, L: lateral

in the developing adult AL. The genetic drivers we identified are invaluable for future studies that may examine the physiology and function of interneurons in the olfactory circuit. In addition, our results will allow for LNs in the *Drosophila* AL to serve as an excellent model of newly born neuron integration into existing mature circuits and may have application across different species.

## Results

**LNs are sequentially recruited to the adult olfactory circuit.** As a first step toward understanding how and when different subtypes of LNs develop and wire into the adult olfactory circuit, we initiated a two-step enhancer trap screen to search for GAL4 drivers that label distinct types of LNs. We started with 1058 InSITE (integrase swappable in vivo targeting element system) GAL4 lines[30] and screened for brains with restricted GAL4 expression in a small number of LNs (Fig. 1a). The initial screen identified 109 candidate LN GAL4 lines, which were then subjected to a second round of screening, wherein brains were co-stained for nuclear *lacZ* or γ-aminobutyric acid[31] to characterize the number of labeled LNs, their morphologies, and their

identities as GABAergic or non-GABAergic neurons. This secondary screen further narrowed down the LN GAL4 lines that distinctly label different subpopulations of LNs in adult brains to 25 candidates (Fig. 1b). Some of the identified GAL4 drivers labeled GABA-negative LNs (Supplementary Table 1), and the neurotransmitter identities of these neurons are currently unknown. Focusing on 15 LN GAL4 lines from the screen and 4 known LN GAL4 lines (*189Y*, *NP3056*, *LCCH3-GAL4*[4], and *krasavietz*[4,16]), we systematically examined the development of distinct subsets of LNs by analyzing their innervation patterns at 9–12 developmental time points (Fig. 2 and Supplementary Fig. 1). We found that a subset of larval LNs was recruited to the adult olfactory circuit during metamorphosis. Later, additional LNs emerged during pupal stages and were recruited to the circuit. We refer to these LNs as larval LNs and adult-specific LNs, respectively. Based on the developmental stage at which labeled LNs emerged, the GAL4 lines were categorized into three groups: (Group 1) GAL4 lines labeling larval LNs and adult-specific LNs (Fig. 2a, Supplementary Fig. 1a and Supplementary Note 1), (Group 2) GAL4 lines labeling adult-specific LNs emerging before 48 h after puparium formation (APF), the time at which cognate olfactory receptor neurons (ORNs) and projection neurons (PNs)

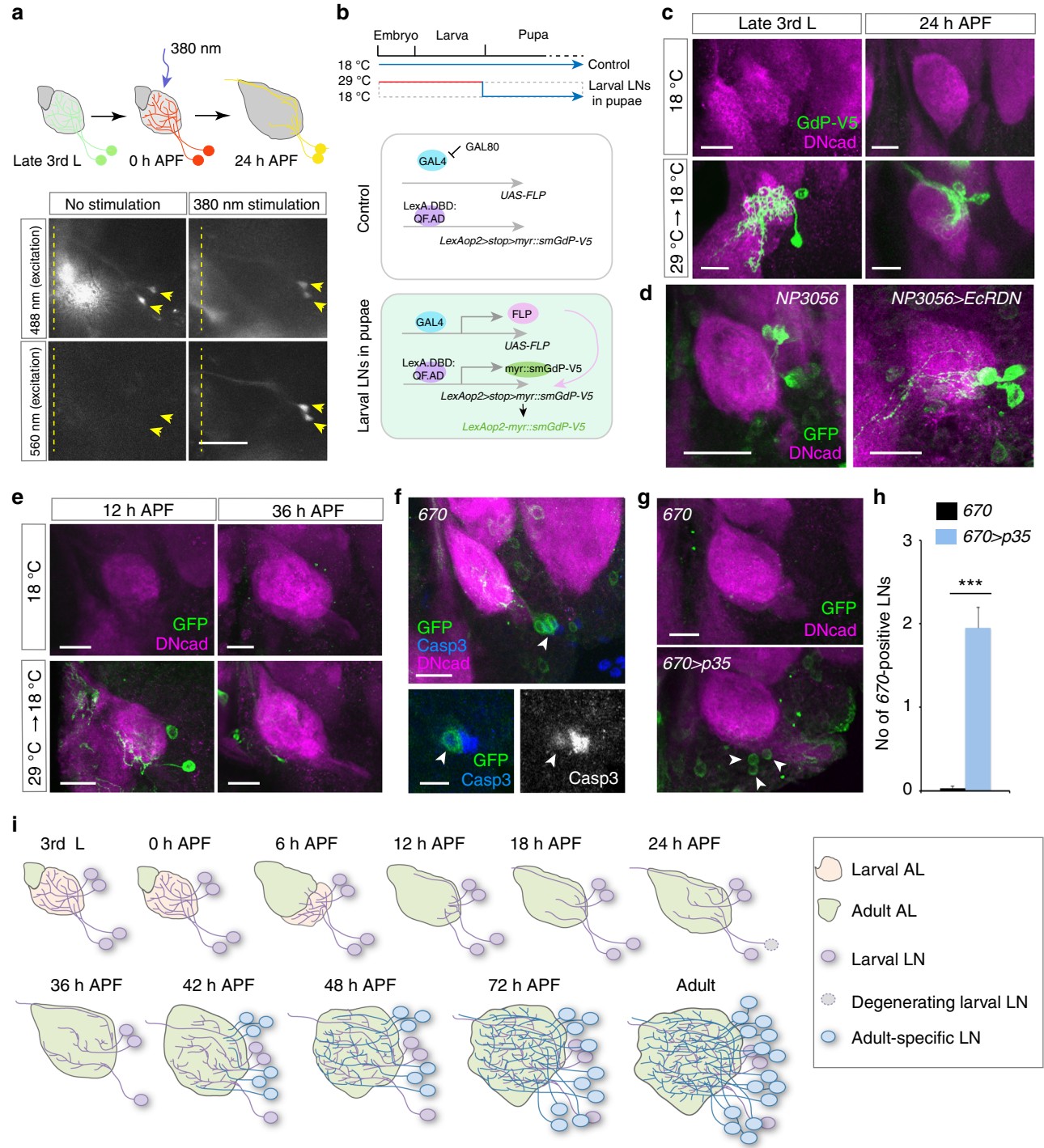

wire together and individual glomeruli emerge (Fig. 2b and Supplementary Fig. 1b), and (Group 3) GAL4 lines labeling adult-specific LNs emerging after 48 h APF (Fig. 2c and Supplementary Fig. 1c). Frequently, LNs in Group 2 and Group 3 were initially observed as a cell soma that lacked processes, suggesting that the late emergence of these LNs is not due to delayed expression of GAL4. The LN GAL4 expression data are also in agreement with previous findings that LNs are sequentially born from embryonic stage to as late as 196 h after egg laying (AEL), which is roughly equivalent to 76 h APF[4,22,32,33] (Supplementary Note 2). Accordingly, these data suggest that LNs in the adult olfactory system are composed of larval LNs, adult-specific LNs recruited during the time at which ORNs and PNs establish their wiring specificity[34], and adult-specific LNs recruited after the completion of such wiring.

**Larval LNs reintegrate into the adult circuit after pruning.** Having observed that distinct types of LNs were sequentially recruited throughout pupal development, we decided to focus our further studies on the recruitment of larval LNs to the adult olfactory circuit. When we closely examined the early development of LNs labeled by Group 1 GAL4 lines, we noticed larval LNs undergo different fates (Fig. 2a). For example, larval LNs

**Fig. 3** Larval LNs undergo pruning or cell death during metamorphosis. **a** (Top) Scheme for Kaede photoactivation. At 24 h APF, stimulated LNs appear with both newly synthesized 488 nm-excited and 560 nm-excited Kaede (yellow). (Bottom) Images from live 24 h APF pupal brains are shown. LNs without 380 nm stimulation only express 488 nm-excited Kaede (100%, $n = 8$ ALs, left panels), while 380 nm-stimulated larval LNs expressed both 488 and 560 nm-excited Kaede (90% analyzed ALs) or only 560 nm-excited Kaede (10%) ($n = 10$ ALs, right panels). Arrowheads indicate LNs. Dashed lines indicate brain midlines. **b** Scheme of two binary systems that together drive myr::smGdP reporter expression exclusively in larval LNs in pupal brains. **c** (Top) Flies raised at 18 °C exhibited no GFP-labeled *NP3056*-LNs in larval and 24 h APF pupal brains ($n = 37$ and 44 ALs, respectively). (Bottom) Flies raised at 29 °C from embryonic stages and then shifted to 18 °C at puparium formation exhibited up to 2 or 3 *NP3056*-positive larval LNs in larval (1.1 ± 0.1 smGdP⁺ LNs, $n = 52$ ALs) and 24 h APF pupal brains (2.0 ± 0.1 smGdP⁺ LNs, $n = 30$ ALs). **d** Compared to 24 h APF control brains ($n = 23$ brains), blocking ecdysone signal in *NP3056*-positive larval LNs leads to pruning failure ($n = 35$ brains). **e** *670*-positive larval LNs. (Top) 12 h APF ($n = 10$ brains) and 36 h APF ($n = 20$ brains) pupal brains of flies raised at 18 °C. (Bottom) 12 h APF ($n = 17$ brains) and 36 h APF ($n = 15$ brains) pupal brains of flies raised at 29 °C from embryonic stages and then shifted to 18 °C at puparium formation. **f** 24 h APF pupal brains were stained for cleaved Caspase 3 and are shown as a projection of stacks (upper panel) or a single confocal section (bottom panels). Arrows indicate *670*-positive larval LNs that express cleaved Caspase 3. **g** 36 h APF pupal brains from control flies ($n = 36$ ALs) and *670 > p35* flies ($n = 20$ ALs). Presumably larval LNs (arrowheads) were only found in *670 > p35* brains. **h** Quantification of presumed larval LN numbers in **g** (mean ± s.e.m.). **i** Model of LN development

labeled by *189Y-GAL4* exhibit extensive trimming of their larval processes (i.e., pruning) between 0 and 12 h APF and re-extended their processes to the developing adult antennal lobe (AL) between 12 and 15 h APF (top row, Fig. 2a). Similarly, *NP3056*-positive larval LNs underwent pruning between 0 and 6 h APF and re-extension of their processes at 12 h APF, while *449*-positive larval LN processes were pruned at 0–12 h APF, with re-extension at 15 h APF (second and third rows, Fig. 2a). Since the re-extended processes of labeled LNs in these three GAL4 lines occupied different domains of the 24 h APF AL, we suspected the three GAL4 drivers may label different subsets of larval LNs. To test this idea, we recombined either any two or all of three GAL4 lines and examined how many LNs were labeled. The results suggested *189Y-GAL4*, *NP3056-GAL4*, and *449-GAL4* label three non-overlapping subsets of larval LNs (Supplementary Fig. 2). These combinations further led us to estimate the minimal number of larval LNs as 26.

To confirm that the labeled LNs, observed in early pupal stages (e.g., 24 h APF), are indeed larval LNs visualized at late 3rd instar larval brains, and not LNs emerging after metamorphosis, we expressed a photoconvertible fluorescence protein, Kaede, in larval LNs. The Kaede protein initially emits fluorescence upon 488 nm excitation, but can be irreversibly altered to emit upon 560 nm excitation by photoconverting the protein with 380-nm UV light[35]. White pupae carrying *NP3056*- or *449*-driven Kaede were subjected to 100 s UV exposure to convert green Kaede to red Kaede, after which the pupae were raised for an additional 24 h (top scheme, Fig. 3a). Indeed, we observed 560 nm-excited Kaede protein, along with newly synthesized 488 nm-excited Kaede protein, in larval LNs within 24 h APF pupal brains (Fig. 3a). A second method combining *tubP-GAL80^{ts36}* with two binary systems, *GAL4-UAS*[37] and *LexA.DBD:QF.AD-LexAop2*[38], was used to confirm the larval LN identities in pupal brains (Fig. 3b). *LexA.DBD:QF.AD* is a fusion protein with the DNA binding domain of LexA and the activation domain of QF. The cross carrying all genetic components was raised at 29 °C from the embryonic stage to inactivate GAL80, thus allowing GAL4 to drive FLP expression. FLP then removed the stop codon between two *FRTs* (>) in the *LexAop2 > stop > myr::smGdP-V5* reporter in larval LNs. myr::smGdP-V5 (also called myr::smGFP-V5) is an epitope V5 tagged non-fluorescent sfGFP protein that is myristoylated, causing the membrane localization of this reporter[39]. Late 3rd instar larvae were then shifted to 18 °C and were maintained at this temperature until analysis at 24 h APF. By this strategy, GAL4 is suppressed by GAL80^{ts} in LNs emerging at pupal stage, and thus, all labeled LNs should be larval LNs. Indeed, we observed 1–3 LNs in *NP3056* pupal brains (Fig. 3c), definitively demonstrating that the labeled LNs we observed in 24 h APF pupal brains are larval LNs per se.

Ecdysone signal regulates global neuronal pruning during metamorphosis[40–45]. We therefore asked whether larval LNs undergo pruning through the conventional ecdysone signaling pathway. Indeed, *NP3056*-positive larval LNs expressed the Ecdysone receptor, EcRB1, at the white pupal stage (Supplementary Fig. 3a). To specifically block ecdysone signal in LNs during pupal metamorphosis and avoid unwanted effects during embryonic and early larval development, we introduced *tubP-GAL80^{ts}* into the flies. With this approach and the substantial half-life of GAL80[ts][36], we were able to restrict the expression of dominant negative Ecdysone receptor (EcR^{Δ655,F654A}, hereafter referred to as EcRDN) in *NP3056*-positive larval LNs from late 3rd instar larval stage to 24 h APF by shifting the larvae to 29 °C from mid-3rd instar larval stage to 24 h APF. When ecdysone signal was disrupted, *NP3056*-positive larval LNs retained processes that extended to the ventral region of the AL and the subesophageal zone (SEZ), reminiscent of their innervation patterns in the late 3rd instar larval brains (Fig. 3d). Similarly, *189Y*-positive and *449*-positive larval LNs also expressed EcRB1 during metamorphosis and underwent ecdysone-mediated pruning (Supplementary Fig. 3a, b).

**A subset of larval LNs undergo apoptosis**. Unlike larval LNs (labeled by *189Y-GAL4*, *NP3056-GAL4* and *449-GAL4*) that undergo pruning during metamorphosis, we noted that another subset of larval LNs, labeled by *670-GAL4*, disappeared after 24 h APF (Fig. 2a). This result could have been due to either cessation of GAL4 expression after 24 h APF or cell death of larval LNs at this stage. Our results from multiple experimental approaches support the idea that *670*-positive larval LNs undergo apoptosis after 24 h APF. First, we introduced *UAS-GAL4* to flies carrying *670-GAL4*, which would sustain GAL4 expression in those larval LNs after the *670-GAL4* was turned off. However, we did not observe any larval LNs in such pupal brains at 36 h APF ($n = 14$ ALs, Supplementary Fig. 3c). Second, we introduced *tubP-GAL80^{ts}*, *UAS-FLP*, *nSyb-QF2w*, and *QUAS > stop > mCD8GFP* to *670-GAL4* flies. Based on a similar idea as was demonstrated in Fig. 3b, this method allows sustained reporter expression in larval LNs. Although *670*-positive larval LNs were observed in 12 h APF pupal brains, we did not observe any LNs in 36 h APF pupal brains from such flies (Fig. 3e). These two sets of experiments ruled out the possibility that GAL4 expression was turned off after 24 h APF in *670*-positive larval LNs. Moreover, we detected cleaved Caspase 3 in these neurons in 24 h APF pupal brains (Fig. 3f). Finally, we tried to rescue larval LNs from cell death by ectopically expressing the viral anti-apoptotic protein, p35[46]. With p35 expression, 2–3 larval LNs were observed in 80% of examined 35 h APF pupal brains (Fig. 3g, h). Notably, we did not

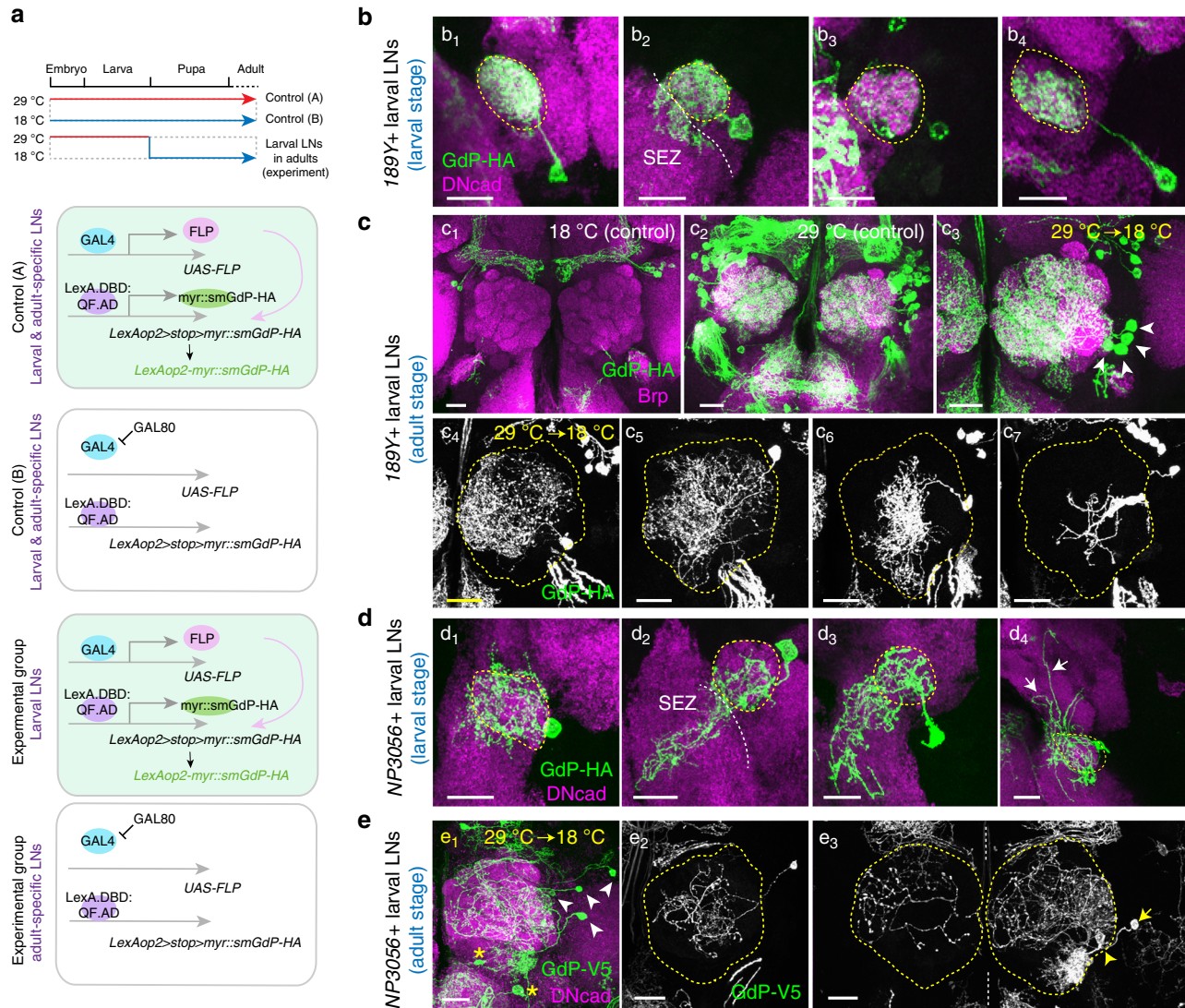

**Fig. 4** Larval LNs acquire different morphologies in adult ALs compared to larval patterns. **a** Scheme of two binary systems that together drive myr::smGdP reporter expression exclusively in larval LNs within adult brains. **b** *189Y*-larval LNs show distinct morphologies in late 3rd instar larval brains. Four types of LNs were observed, as shown in $b_1$ ($n = 22/68$ SCCs), $b_2$, ($n = 30/68$ SCCs), $b_3$ ($n = 9/68$ SCCs), and $b_4$ ($n = 7/68$ SCCs). Supplementary Note 3 contains detailed descriptions of larval LNs in **b** and **d**. Yellow and white dashed lines contour ALs and SEZ, respectively. **c** The morphologies of *189Y*-positive larval LNs in adult brains. $c_1$ Negative control flies were maintained at 18 °C throughout development, and no smGdP-labeled LNs were observed in adult brains ($n = 43$ brains). $c_2$ When flies were raised at 29 °C throughout development, both larval and adult-specific LNs were labeled ($n = 26$ brains). $c_3$–$c_7$ When flies were raised at 29 °C from embryonic stage and shifted to 18 °C at puparium formation, as many as four LNs were observed in the adult brains (arrowheads, $n = 17$ brains) ($c_3$). These LNs may have processes in dorso-medial glomeruli ($c_4$) or central glomeruli ($c_5$). Otherwise they may pack densely in a small subset of central glomeruli ($c_6$) or sparsely innervate multiple glomeruli ($c_7$). ALs are contoured by yellow dashed lines. **d** *NP3056*-larval LNs show distinct morphologies in late 3rd instar larval brains. The four types of LNs are shown in $d_1$ ($n = 21/86$ SCCs), $d_2$ ($n = 24/86$ SCCs), $d_3$ ($n = 25/86$ SCCs), and $d_4$ ($n = 16/86$ SCCs). Arrows indicate LN processes in dorsal brain regions. **e** Similar to **c** but for *NP3056*-GAL4. When flies were shifted to 18 °C at puparium formation, as many as four LNs were observed in adult brain (arrowheads, $n = 54$ brains) ($e_1$). These LNs may have processes that cover the central AL ($e_2$), innervate both ALs ($e_3$, arrow, bilateral LN), or the neuron may innervate VP1, VP2, and VP3 ($e_3$, arrowhead, oligo-glomerular LN). Asterisks denote neurons that are not LNs

observe dense LN processes in the AL, suggesting ectopically expressed p35 may rescue cell death of the soma and primary neurites but not the degeneration of processes. These data conclusively demonstrated that *670*-positive larval LNs normally undergo apoptosis after 24 h APF. Our findings support a model that LNs in the adult olfactory system compose a subset of larval LNs and adult-specific LNs, which are sequentially recruited to the developing adult AL as (1) stage 1 (0–12 h APF): larval LNs undergo pruning; (2) stage 2 (12–48 h APF): larval LNs either re-innervate the AL or die, and adult-specific LNs innervate the AL,

and (3) stage 3 (48 h APF-eclosion): after individual glomeruli are formed and PN-ORN wiring and synapse formation are finished, some adult-specific LNs sequentially innervate the AL (Fig. 3i).

**Larval LNs acquire different morphologies in adult brains.** Since we had observed that a subset of larval LNs is maintained in pupal brains, we next asked whether those larval LNs eventually die at the late pupal stage and if not, whether they transform into different morphologies in adult brains. We began to answer these

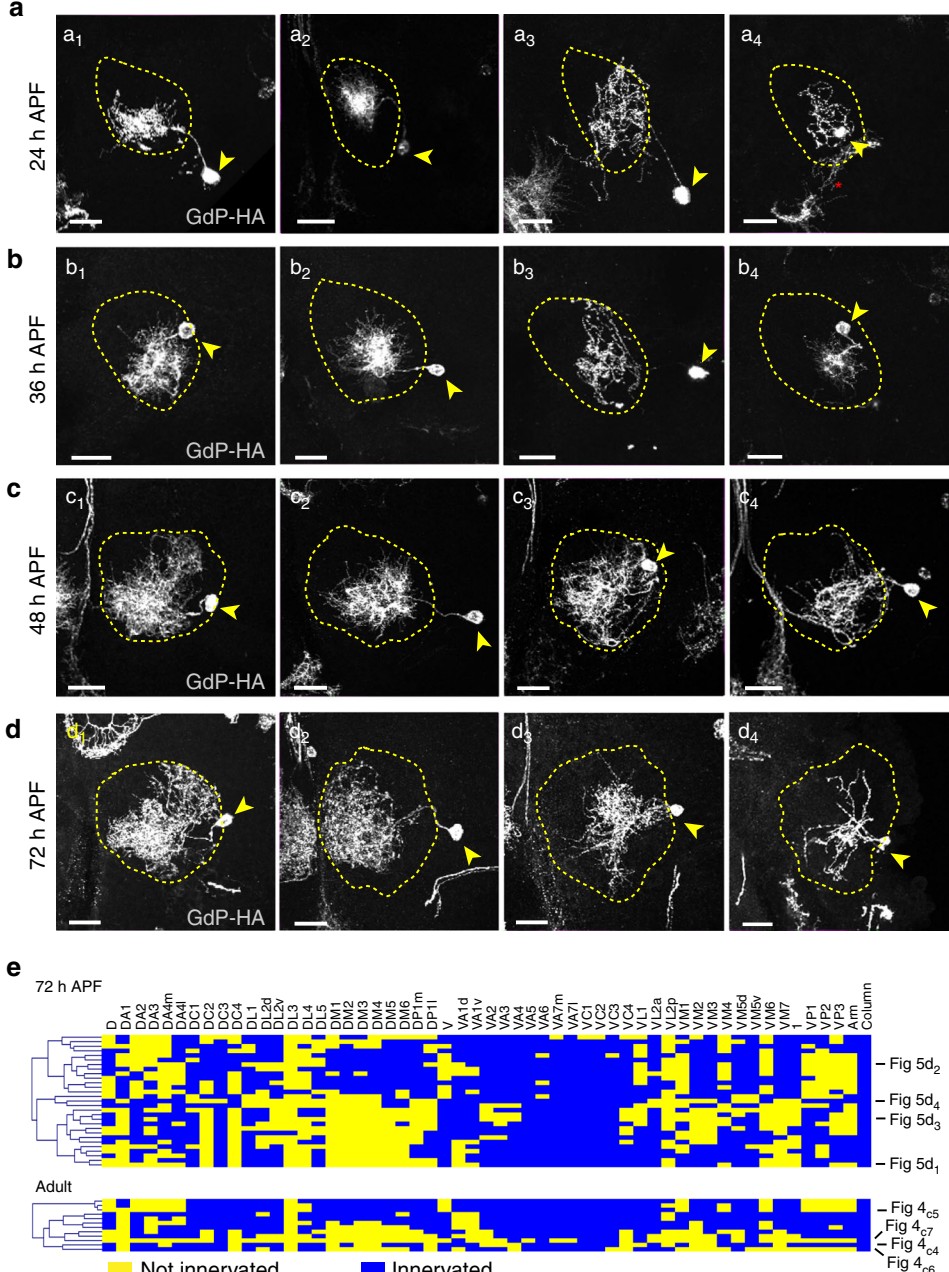

**Fig. 5** Morphogenesis of 189-positive larval LNs during pupal development. **a–d** The same labeling strategy as in Fig. 4c was used to examine the morphologies of *189Y*-positive larval LNs in 24 h APF ($n = 50$ SCs/154 ALs) **a**, 36 h APF ($n = 51$ SCs/211 ALs) **b**, 48 h APF ($n = 51$ SCs/138 ALs) **c** and 72 h APF ($n = 32$ SCs/121 ALs) **d** pupal brains. Brains were stained with HA and DNcad (not shown) to visualize neurons and neuropil, respectively. The larval LNs were observed to have processes that densely covered the medioventral region and the dorsolateral region of the AL ($a_1$–$d_1$), that are mostly concentrated in the medio-central portion of the AL ($a_2$–$d_2$), that innervate the centrolateral portion of the AL ($a_3$–$d_3$), or that are sparse and irregular in the AL ($a_4$–$d_4$). LNs from each developmental stage were categorized by the similarity of their neurite density, coverage, and occupied regions. LNs shown in $a_1$–$d_1$, $a_2$–$d_2$, $a_3$–$d_3$, and $a_4$–$d_4$ are likely, but not necessarily, of the same type. ALs are contoured by yellow dashed lines. Yellow arrowheads indicate LN soma. Asterisk denotes processes from non-LN cells that are not in the AL. Scale bars, 20 μm. **e** Hierarchical clustering of the innervation profiles of individual LNs in 72 h APF pupal brains ($n = 29$) (top) or adult brains ($n = 12$) (bottom). These LNs showed heterogeneous innervation patterns, but fall into four broad types

questions by first profiling the identities (morphologies) of several types of larval LNs. We found that *189Y-GAL4* labeled diverse larval LNs with distinct morphologies in late 3rd instar larvae (Fig. 4b), which is in agreement with previous findings[29] (Supplementary Note 3). We then asked whether those larval LNs die at later pupal stages and if not, what their identities (morphologies) are in adult brains. Since these GAL4 drivers label additional adult-specific LNs at later pupal stages (Fig. 2a and

Supplementary Fig. 1a), it is difficult to address these questions by conventional labeling and clone analysis methods, such as MARCM[47]. To overcome this issue, we used similar methods as described in Fig. 3b, wherein we combined *tubP-GAL80^ts* with two binary systems, *GAL4-UAS* and *LexA:QF-LexAop2*. The crosses carrying all genetic components were raised at 29 ℃, beginning from the embryonic stage to inactivate GAL80. These flies were then shifted to 18 ℃ once they entered the white pupal

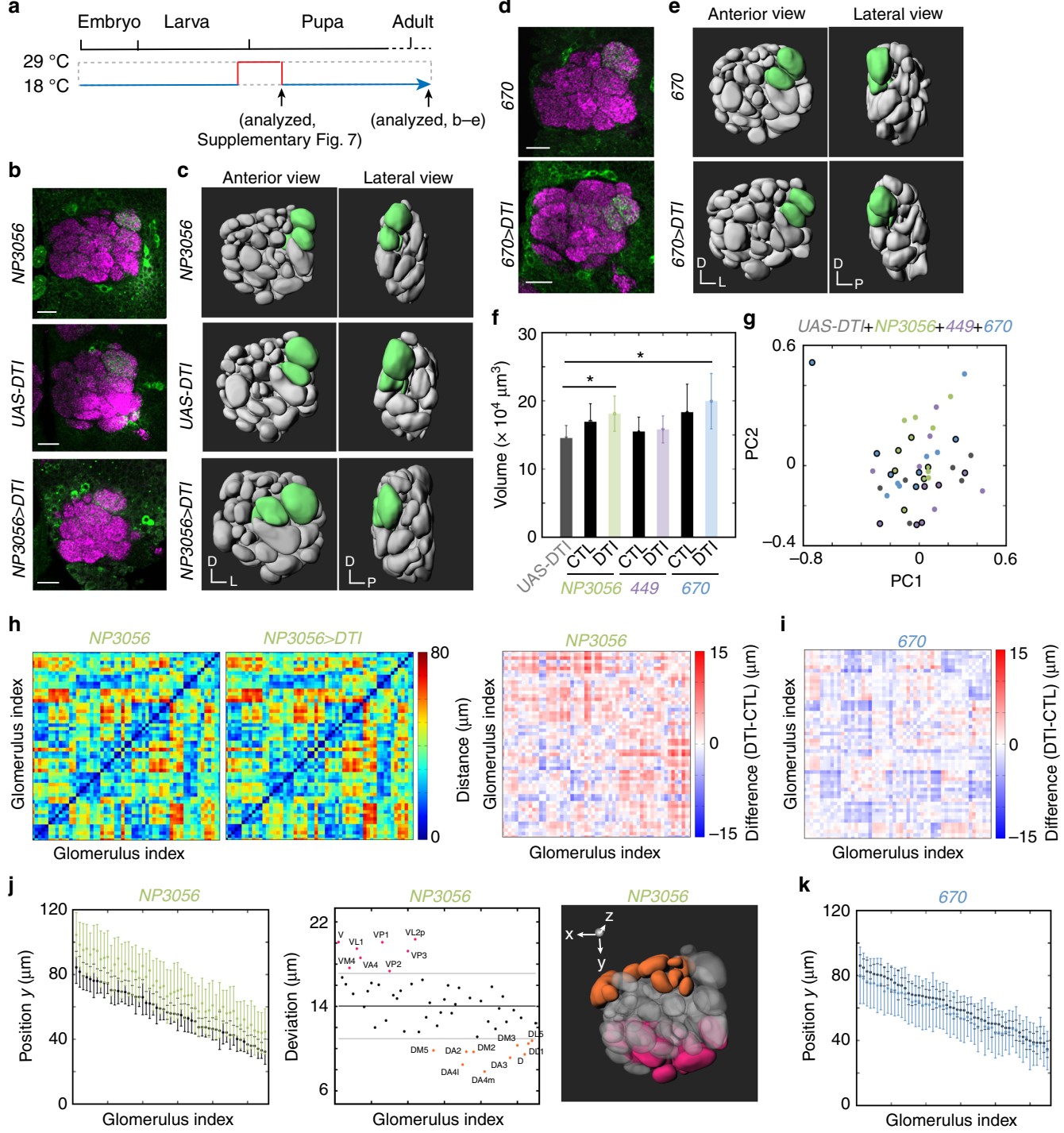

stage and were maintained at this temperature until they became adults (Fig. 4a). By this strategy, GAL4 is suppressed by GAL80[ts] in adult-specific LNs, and as such, all labeled LNs in adult brains are larval LNs. To demonstrate the reliability of this system, we raised the crosses at either 18 °C or 29 °C throughout development, and adult brains showed either no labeled LNs or all LNs were labeled (Fig. 4c$_1$, c$_2$). Interestingly, 189Y-positive larval LNs exhibit different innervation patterns in adult brains of the experimental group (29 °C to 18 °C). At most, we found four LNs in one brain hemisphere (Fig. 4c$_3$). In those brains carrying single LNs, we identified four types of LNs: three regional LNs (Fig. 4c$_4$, c$_5$, c$_6$) and an LN sparsely innervating multiple glomeruli

(Fig. 4c$_7$). We further examined how 189Y-positive larval LNs reintegrate into developing adult ALs and establish distinct morphologies through pupal development. Our results suggested that distinct subtypes of 189Y-positive larval LNs are likely to already have distinct innervation patterns at the initial stage of neurite re-extension and use different strategies to expand their neurite territories (Fig. 5).

Similarly, NP3056-positive larval LNs show diverse innervation patterns in larval ALs (Fig. 4d, Supplementary Note 3) and have distinct innervation patterns in adult brains (Fig. 4e). Also similar to the 189Y-positive cells, we found up to four NP3056-positive LNs in one adult brain hemisphere (Fig. 4e$_1$). In those brains

**Fig. 6** Genetic ablation of larval LNs causes aberrant glomeruli organization. **a** Schematic of temperature shift. Larvae were transiently raised at 29 °C from late 3rd instar larvae to 6.5 h APF. **b, d** PN dendrites targeting to glomeruli DA1, VA1d, and DC3 of control and *NP3056 > DTI* (**b**), and control and *670 > DTI* (**d**) brains. Adult brains were stained for Bruchpilot (magenta) to visualize individual glomeruli and GFP to visualize PN dendrites (green). **c, e** 3D reconstruction of ALs from control (top) and *LN > DTI* brains were shown in anterior view (left) and lateral view (right), respectively. Glomeruli DA1, VA1d, and DC3 are in green. **f** Bar graph showing the volume of *UAS-DTI* control, *LN-GAL4* control, or *LN > DTI* ALs. **g** Ten variables (Methods) extracted from *UAS-DTI* control (gray dots), *LN-GAL4* control (green (*NP3056*), magenta (*449*) or blue (*670*) dots outlined with black), and *LN > DTI* (green (*NP3056*), magenta (*449*), or blue (*670*) dots) ALs were subjected to principle component analysis (PCA). No obvious separation of control and *LN > DTI* ALs was observed. **h** The relative distance between any two glomeruli in *NP3056-GAL4* control (left panel) and *NP3056 > DTI* (middle panel) ALs. The differences between a given pair of glomeruli in control and *NP3056 > DTI* ALs are shown in the right panel. **i** The differences between a given pair of glomeruli in control and *670 > DTI* ALs. **j** (Left) The *y*-position of individual glomeruli in *NP3056* control and *NP3056 > DTI* ALs. Glomeruli were aligned based on their mean *y*-position value in the *GAL4* control ALs (Supplementary Note 4). Note that *NP3056 > DTI* glomeruli have larger *y*-position values. (Middle) The relative shift of individual glomeruli in *NP3056 > DTI* ALs was quantified as the deviation. Glomeruli were aligned as in the left panel. The mean deviation of all glomeruli is 14.0318 (black line). Gray lines indicate one standard deviation (SD) from the mean, which contains 68.3% of the glomeruli. (Right) A 3D-reconstructed AL shows glomeruli with deviation > 1 SD (pink) and < −1 SD (orange). **k** The *y*-position of individual glomeruli in *670* control and *670 > DTI* ALs is shown. No obvious *y*-position shift was observed in *670 > DTI* ALs. **f**, **j**, **k** Mean ± s.d

carrying 1–2 LNs, we identified three types of LNs: a regional LN (Fig. 4e₂), two bilateral LNs (arrow, Fig. 4e₃, as one example), and an oligo-glomerular LN which innervates glomeruli VP1, VP2, and VP3 (arrowhead, Fig. 4e₃). These data are in agreement with a previous MARCM analysis of *NP3056*-positive LNs, which showed that these same three types of LNs were born at the embryonic stage[4]. Interestingly, the bilateral LN first innervates posterior glomeruli in ipsilateral and contralateral ALs and later innervates the anteroventral glomeruli in both ALs (Supplementary Fig. 4). Similar experiments were performed with *449-GAL4* and *449*-positive larval LNs; 3–24 LNs per AL were observed in adult brains (*n* = 51 brains, Supplementary Fig. 5). Therefore, our data not only confirmed that *189Y*-positive, *NP3056*-positive, and *449*-positive LNs in 24 h APF pupal brains are bona fide larval LNs, but also suggested that these larval LNs undergo pruning and reintegration into the adult olfactory circuit. Although the current experimental approaches did not allow us to match individual larval LNs with their adult identities, a striking difference in innervation patterns between larval and adult stages was clearly observed.

**Larval LNs shape the global organization of developing AL.** The re-extended processes of *189Y*-positive, *NP3056*-positive, and *449*-positive larval LNs respectively occupied the medial, lateral and central regions of the developing AL at 18–24 h APF (Fig. 2a). Furthermore, the *NP3056*-positive and *449*-positive re-extended neurites, in addition to *670*-positive degenerating neurites, differentially and partially overlapped with PN dendrites in the developing AL (Supplementary Fig. 6). Therefore, we hypothesized that the regional occupation of LN processes may serve as landmarks for the developing adult AL, which instruct the targeting of PN dendrites and/or ORN axons. We then predicted that ablating different subsets of larval LNs should lead to morphological changes that may be reflected by mistargeting of PN dendrites. To test this idea, we combined *tub-GAL80^ts* and toxin *UAS-DTI* to allow *NP3056*-driven DTI to be expressed only from late 3rd instar to 6.5 h APF. We chose *NP3056* because it has stronger GAL4 expression than *189Y* in LNs at early pupal stage. Such transient DTI expression effectively ablated *NP3056*-positive larval LNs (Supplementary Fig. 7b, c) but did not lead to obvious mistargeting phenotypes of PN dendrites (Fig. 6b, Supplementary Table 2). Interestingly, 3D reconstructions of glomeruli and ALs revealed distortions in the *NP3056*-driven DTI brains (Fig. 6c, Supplementary Fig. 10). The total volume and analyzed variables of the AL did not show obvious differences (Fig. 6f, g). However, the relative distance between any two glomeruli was altered in the *NP3056*-driven DTI brains (Fig. 6h). Analysis of the relative positions of glomeruli in the *Y*-axis demonstrated that a group of

dorsal glomeruli were shifted posterodorsally (orange, Fig. 6j, Supplementary Fig. 8c, e–h, Supplementary Table 4) and a second group of ventral glomeruli were shifted anteroventrally (pink, Fig. 6j, Supplementary Fig. 8e–h). These results suggest that *NP3056*-positive larval LNs may contribute to globally shaping the dorsal–ventral (D–V) and antero-posterior (A-P) axes of glomeruli within the AL and locally determine the relative position of individual glomeruli (Supplementary Fig. 12).

We then asked what would happen to the AL organization when a larger group of larval LNs were ablated by *449-GAL4*-driven DTI. Similar to the *NP3056* LN driver, we did not observe obvious mistargeting of PN dendrites in *449*-positive larval LN ablated brains (Supplementary Figs. 7d, e, 9b, c, Supplementary Table 2). However, this GAL4 line, without ablation, showed a glomerular shift along the *Y*-axis (Supplementary Fig. 9d), making the effect of ablating *449*-positive larval LNs difficult to interpret (Supplementary Fig. 9d, Supplementary Table 4).

**Ablating dying larval LNs has minimal effects on AL geometry.** Intriguingly, the neurites of *670*-positive larval LNs remained in the ventral part of the developing adult AL at 24 h APF, a time at which the axons of all larval ORNs have been fully eliminated[48] (Fig. 2). Based on this observation, we asked whether this delayed degeneration of larval LN neurites maintains the ventral identity of the developing AL, prior to the arrival of adult ORN axons. If so, we would expect that prematurely ablating *670*-positive larval LNs may confuse the targeting of PN dendrites and/or adult ORN axons. To test this idea, we induced expression of *670-GAL4*-driven DTI from late 3rd instar to 6.5 h APF (Fig. 6a). Transient DTI expression effectively ablated *670*-positive larval LNs (Supplementary Fig. 7f, g), however, PN dendrites still correctly targeted to DA1, VA1d, and DC3 (Fig. 6d, Supplementary Table 2). Although we observed a mild change in the relative distance between any two glomeruli (Fig. 6i), no clear shift of glomerular positions along the *Y*-, *X*-, or *Z*-axis in the *670 > DTI* brain (Fig. 6k, Supplementary Fig. 8d, Supplementary Table 4) was detected. Taken together, the results from ablating *NP3056*-, *449*-, and *670*-positive LNs imply that the spatial distributions of a combination of processes from different subsets of larval LNs may participate in shaping the global organization of the AL.

## Discussion

Systematic explorations into how distinct types of interneurons are specified, morphologically differentiate, and properly wire into neuronal circuits are crucial for understanding how malfunction of neuronal circuits arises. With a degree of complexity that is sufficient to model circuits in higher organisms, diverse

LNs of the *Drosophila* olfactory system offer an unparalleled opportunity to conduct such studies. Here, we demonstrate that LNs in the adult olfactory circuit are composed of both larval LNs and adult-specific LNs. Sequential recruitment of LNs to the adult olfactory circuit occurs both before and after the time when ORNs and PNs establish their wiring specificity and the emergence of glomeruli. Focusing on subpopulations of larval LNs, we found that these cells are selectively integrated into the adult circuit. Some larval LNs prune and reintegrate to the developing circuit, while others die. The processes of reintegration and death in larval LNs are temporally coordinated and appear to serve as a framework that supports and/or maintains the global organization of the developing AL.

Based on the results from our systematic analysis of emergence times and morphologies of neurons labeled by 19 GAL4 drivers, we propose three waves of LN emergence in the adult olfactory circuit (Fig. 3i). First, during metamorphosis, a subset of larval LNs are pruned and reintegrated into the developing adult AL, while others die. Second, some adult-specific LNs innervate the AL during the period when PN dendrites and ORN axons actively form wiring contacts (roughly 24–48 h APF). Third, a subset of adult-specific LNs appears after cognate ORNs and PNs establish their connections (48 h APF-eclosion). The proposed model and identified GAL4 drivers may serve as a blueprint for future developmental and functional studies of distinct types of LNs or the mechanisms that regulate the integration of adult-born neurons.

We noted that as soon as the labeled LNs emerge, only the soma and primary branches can be observed in most cases. This pattern of soluble fluorescent signal implies that the neurons were initially observed at the early targeting stage (Fig. 2, Supplementary Fig. 1). In addition, previous findings have shown that LNs are born from embryonic stage to as late as 196 h AEL (~76 h APF)[4,22,32,33]. It is possible that the GAL4 expression is delayed in LNs of Group 2 and Group 3. As such, labeled LNs would be born and integrate to the circuit at early developmental stages but would not be observed until GAL4 was expressed at later developmental stages. Although this possibility cannot be fully excluded by our current data, our observations strongly support the idea that some adult LNs are born and integrate into the circuit at early pupal stages (before 48 h APF), while others are born and integrate at late pupal stages (after 48 h APF) (Supplementary Note 2).

Axonal pruning is one of the strategies that is widely implemented in vertebrate and invertebrate nervous systems to achieve the final wiring of neural circuits[49,50]. The *Drosophila* adult olfactory system is not built from the ground up. Instead, portions of the larval olfactory system are repurposed and integrated into the new structures. For example, larval projection neurons (PNs) and γ Kenyon cells in the mushroom body (MB) undergo pruning and reintegration into the developing adult olfactory circuit during the larva-to-adult transition[41,43]. In contrast to these cells, all larval olfactory sensory neurons die[48]. Yet the fate of larval LNs during metamorphosis has not been previously described.

Here we estimate the lower-bound number of larval LNs is 26 in late 3rd instar larval brains (Supplementary Fig. 2). Among this set of larval LNs, different populations take on different fates during metamorphosis. Some of the neurons prune and reintegrate to the adult olfactory circuit, while others undergo apoptosis (Figs. 2a, 3, Supplementary Figs. 1a, 3). Previous studies have examined the mechanisms that control pruning, reintegration and apoptosis in *Drosophila* neurons. For example, transient calcium activity is known to trigger dendritic arborization neuron pruning[51] and nitric oxide is involved in the switch between MB γ neuron pruning and re-extension[44]. It will be of interest to test

whether either of these mechanisms regulates pruning and neurite re-extension in LNs.

Pruning larval LNs are likely to acquire innervation identities at the initial stage of neurite re-extension (Fig. 5, Supplementary Fig. 4). These LNs may initially target the center of an AL with subsequent neurite spreading (Fig. 5a$_2$–d$_2$) or they may innervate a particular region of the AL, followed by innervation of a second subset of glomeruli (Fig. 5a$_1$–d$_1$, Supplementary Fig. 4a$_1$–d$_1$). Thus, our identification of larval LN markers with specific fates provides an excellent platform to study mechanisms of interneuron degeneration and regrowth under normal physiological and pathological conditions. However, the method that we used to probe single larval LNs in different pupal stages is not sufficient to unambiguously trace the development of single larval LNs through pupal stages. Several groups have successfully cultured larval or pupal brains[44,52–55] for a long time periods. In future studies, these live-imaging methods will be highly applicable for monitoring larval LN development and allow further delineation of LN subsets.

Tremendous efforts have been devoted to understanding the mechanisms that instruct the targeting and matching of PN dendrites and ORN axons to form individual glomeruli. Yet it is still unclear why individual glomeruli are located in stereotypical positions within the AL. We identified four subsets of larval LNs with re-extended or degenerating processes that occupy distinct domains of the 24 h APF AL (Fig. 2a). When the innervation of neurites from three subtypes were individually disrupted, the targeting of PN dendrites appeared to be unaffected (Supplementary Table 2), suggesting that PN dendrite targeting is independent of larval LN integration to the adult AL. However, when *NP3056*-positive larval LNs were ablated, the stereotypical glomerular positioning was disturbed, as evidenced by the relative distance between any two glomeruli and differential shifts of distinct groups of glomeruli along the Y-axis (Fig. 6, Supplementary Figs. 7, 8, 9). These two phenotypes have no obvious correlation (Supplementary Fig. 12d). Our findings suggest that larval LNs may serve as a global framework to instruct the localization of individual glomeruli in the developing adult AL. Although we did not observe dendrite mistargeting in PNs, whether such glomerular disorganization would affect the connectivity among LNs is an open question. In the future, it will be interesting to test whether the neurons need to be functional, or if they simply act as a scaffold to maintain the tissue shape during metamorphosis.

It is not clear how distinct types of olfactory LNs acquire their diversity and variability. We found that through pruning and re-extension, larval LNs reintegrate into adult circuits with morphologies that are different from those exhibited in larval stages (Fig. 4). It is also currently unknown why larval LNs undergo such dramatic morphological changes, however, our results strongly suggest that LNs exhibit extremely high plasticity and are likely to execute complicated, embedded genetic programs with precise temporal control. A subset of Fru-expressing interneurons exhibits different developmental programs that lead to distinct neurite morphologies in males and females[56]. It will be of interest to examine whether the heterogeneous innervation patterns we observed reflect LN variability or dynamic neurite innervation during development (Fig. 5e) and whether sexual dimorphism is one of the mechanisms underlying such heterogeneity.

Some LNs innervate the AL after the wiring and synapse formation between cognate PNs and ORNs are completed (Fig. 2c and Supplementary Fig. 1c). Adult neurogenesis significantly contributes to the plasticity and complexity of vertebrate neural circuits, including the olfactory system[57]. However, it is still not clear whether the integration of adult-born neurons into a mature

circuit would involve competition for existing synapses or have any consequences on existing synapses. For studies directed toward this question and others, the fly olfactory interneurons may serve as an excellent model system to elucidate underlying mechanisms of LN integration into established circuits.

Overall, our study provides a blueprint for LN emergence and integration into the adult olfactory circuit. By identifying specific genetic drivers that define LN subpopulations and characterizing the temporal fate of these cells, we have established a novel toolkit for studies of LN function and dysfunction. These future studies may be directed toward discovering mechanisms that govern LN apoptosis, pruning, and reintegration, or integration into developing or mature circuits.

## Methods

**Fly care and genotypes**. Flies were raised at 25 °C with 12 h light–dark cycles, unless otherwise noted for temperature shift experiments. Except for the InSITE GAL4 screen (Fig. 1) that used only female brains, male and female brains were not separately analyzed in the other experiments. $w^{1118}$ (BL-5905), 189Y-GAL4[58] (BL-30817), NP3056-GAL4[4], krasavietz-GAL4[59], LCCH3-GAL4[4], c739-GAL4[60], nSyb-LexA.DBD::QF.AD (attP2)[38], nSyb-QF2w.P (attP2)[38] (BL-51960), QUAS(FRT.stop)mCD8GFP.P (10)[61] (BL-30134), tubP-GAL80[ts] (2)[36] (BL-7017), tubP-GAL80[ts] (20)[36] (BL-7019), tubP-GAL80[ts] (7)[36] (BL7018), UAS-nuclacZ, UAS-mCD8GFP[47], UAS-EcR.B1-ΔC655,F645A (TP1)[62] (BL6869), UAS-Kaede.A (3) (BL-26161), UAS-FLP (II), 13xLexAop2(FRT.stop)myr::smGdP-V5 (attP40)[39] (BL-62107), 13xLexAop2(FRT.stop)myr::smGdP-HA (VK00005)[39] (BL-62106), QUAS(FRT.stop)mCD8-GFP.P (10)[61] (BL-30134), UAS-GAL4.H (12B)[63], UAS-p35.H (BH3)[64] (BL-6298), ey3.5-GAL80[65], GH146-QF[61], QUAS-mtdTomato-3xHA[61], MZ19-mCD8GFP(y +)[66], and UAS-DTI[67] were used. Detailed genotypes of flies used in individual experiments are listed in Supplementary Table 5.

**Immunohistochemistry**. Larval, pupal or adult brains were dissected in freshly prepared 4% paraformaldehyde/PBS and fixed at room temperature (RT) for 20 min. The samples were then washed for 20 min, three times with PBST (0.3% Triton X-100, PBS) at RT, and blocked with 5% normal goat serum (NGS)/PBST (005-000-121, Jackson ImmunoResearch Laboratories) at RT, 30 min. Brains were incubated with diluted primary antibodies in 5% NGS/PBST at 4 °C for 2–3 d, followed by three 20 min PBST washes at RT. Tissues were then incubated with diluted secondary antibodies in PBST at 4 °C overnight and then subjected to three 20 min PBST washes at RT. The samples were then soaked in SlowFade™ Gold Antifade Mountant (S36936, Invitrogen) and mounted in the same solution with support of coverglass (for adult brains) or silicon gel (for larval and pupal brains).

Primary antibodies used were mouse anti-Brp (nc82, DSHB; 1:30), rat anti-mCD8 (MCD800, Invitrogen; 1:100), rat anti-DN-cadherin (DN-Ex#8, DSHB; 1:25), rabbit anti-β-galactosidase (559762, CAPPEL/MP Biomedicals; 1:1000), mouse anti-V5 (11417489, Invitrogen; 1:150), rabbit anti-cleaved Caspase 3 (Cat No 9661, Cell Signaling; 1:100), rabbit anti-GABA (A2052, Sigma; 1:200), rat anti-HA (3F10, 11867423001, Roche; 1:200), mouse anti-EcRB1 (4D.1, DSHB, concentrated; 1:50), rabbit anti-HA (ab9110, Abcam; 1:500-1:1000), mouse anti-GFP (3E6, A-11120, Invitrogen; 1:1000), rabbit anti-GFP (A-6455, Invitrogen; 1:1000).

Secondary antibodies used were goat anti-mouse IgG, goat anti-rat IgG and goat anti-Rabbit IgG which are conjugated with Alexa 488, DyLight 488, Cy3, Cy5, or DyLight 649. All secondary antibodies were from Jackson ImmunoResearch Laboratories or Invitrogen and used at 1:500 (Supplementary Table 6).

Brains were imaged using a Zeiss LSM780 equipped with Mai-Tai HP-1040 (Spectra-Physics) or a Zeiss LSM700 confocal microscope and processed using Zen black (Carl Zeiss), Fiji [https://fiji.sc], and Adobe Photoshop.

**Pupal staging and developmental study**. White pupae (defined as 0 h APF) were collected and aged at 25 °C unless otherwise indicated. For developmental studies (Fig. 2 and Supplementary Fig. 1), 5–17 brains from individual GAL4 lines were analyzed at each time point.

**InSITE screening**. A collection of 1058 GAL4 lines were subjected to the first round of screening which was conducted by crossing individual IS-GAL4 to UAS-mCD8GFP flies (both are on the Oregon-R isogeneic background). At least five brains of 2–4 old females were immunostained for Bruchpilot and mCD8GFP to examine the GAL4 expression patterns in brains. In the second round of screen, 109 GAL4 lines with expression in subsets of LNs were outcrossed to flies carrying UAS-nuclacZ, UAS-mCD8GFP. Two to four-day-old female brains were stained with β-Galactosidase, GABA, mCD8GFP, and Bruchpilot to further characterize the number and identities of labeled LNs.

**Tracing larval LNs with Kaede**. The crosses were raised in constant darkness. Intact white pupae carrying NP3056- or 449-driven Kaede were subjected to 380

nm stimulation for 15, 50, or 100 s (CFI Plan Achromat 10×, C-LHGFI HG LAMP (130 W, 100% power), C-FL Epi-FL Block DAPI, Nikon Ni-E microscope), and aged to 24 h APF at 25 °C. Pupal brains were dissected in 5% $CO_2$, 95% $O_2$-prebubbled adult-like hemolymph (108 mM NaCl, 5 mM KCl, 5 mM Hepes, 5 mM Trehalose, 5 mM sucrose, 26 mM $NaHCO_3$, 1 mM $NaH_2PO_4$, 2 mM $CaCl_2$, 1 mM $MgCl_2$, 275 mOsm osmolarity, pH 7.3)[68] and observed by live imaging under 480 nm and 560 nm excitation through Colibri2 (Ilumina) and EMCCD camera Rolera em-c2 (Qimaging) equipped on a LSM700. Images were analyzed with Zen blue. Brains with 15, 50, and 100 s stimulation at 380 nm showed similar results and were thus pooled for analysis.

**Single-cell clones of larval LNs**. Larvae carrying the heat-shock flipase hs-FLPG5.PEST, a flip-out reporter (10xUAS-FRT-stop-FRT-myr::smGdP-HA) and 189Y-GAL4 or NP3056-GAL4 were subjected to a 30 min 37 °C heat-shock at age 48–104 h after egg laying, to remove the stop site in the flip-out reporter.

**Larval LNs in pupal and adult brains**. Female virgins were maintained at 18 °C for at least one night before the crossing. When the crosses were raised at 29 °C throughout development (control A, Fig. 4a), GAL80 was suppressed, allowing GAL4 to drive FLP expression. Under these conditions, FLP will bind to the two FRT sites of the flip-out reporter and remove the stop site, to produce a functional LexAop-myr::smGdP-HA expression construct. The pan-neuronal driver nsyb-LexA.DBD::QF:AD can bind to LexAop2 and drive myr::smGdP-HA expression. When the crosses were constantly raised at 18 °C (control B, Fig. 4a), ubiquitously expressed GAL80 suppressed GAL4 activity and therefore the flip-out reporter remained nonfunctional. To induce reporter expression in larval LNs only, the crosses were raised at 29 °C as soon as they were established. Late 3rd instar larvae from those crosses were picked and kept at 18 °C until analyzed pupal stages or eclosion. By this strategy, only those neurons with GAL4 expression before the end of the larval stage carried functional LexAop2-myr::smGdP-HA (larval LNs in experimental group, Fig. 4a). Since GAL80 expression was resumed after entering the pupal stage, GAL4 became suppressed, and neurons which did not express GAL4 in the pupal stage did not contain a functional reporter (adult-specific LNs in experimental group, Fig. 4a). For NP3056- and 449-positive larval LNs, the reporter LexAop2 > stop > myr::smGdP-V5 was used. For 670-positive larval LNs, nsyb-QF2w and QUAS > stop > mCD8GFP was used.

**Hierarchical clustering of LN innervation profiles**. The innervation patterns of 189Y single cells were analyzed from confocal stacks. Single cells with sparse ORN axons were excluded. Glomeruli that were innervated or not innervated by a given LN were scored as 1 and 0, respectively. The binary innervation profiles of LNs were subjected to hierarchical clustering with optimized gene leaf order, Pearson correlation, and complete linkage clustering (MultiExperiment Viewer (MeV) 4.6.1 [http://mev.tm4.org])[31].

**Genetic ablation of NP3056-, 449-, or 670-positive larval LNs**. ey-GAL80 was introduced to the flies to suppress GAL4-expression in ORNs, if any, during development. Control and experimental crosses were raised at 18 °C to allow GAL80[ts] to function, and were transferred to 29 °C at the late 3rd instar larval stage to inactivate GAL80[ts] and thus allow DTI expression in GAL4-positive larval LNs. White pupae (0 h APF) were picked and kept at 29 °C for an additional 5 h (equivalent to 6.5 h APF at 25 °C). The pupae were either dissected and immunostained to examine the ablation efficiency (Supplementary Fig. 7) or transferred back to 18 °C to stop DTI expression. 0-5 d after eclosion, adult brains were dissected and immunostained to visualize the innervation of PN dendrites and antennal lobe organization (Fig. 6, Supplementary Fig. 9). Due to the substantial half-life of GAL80[ts][36], DTI is likely to be expressed from wondering larval stage/early metamorphosis to about 12 h APF.

**PN dendrite targeting in adult ALs with ablated larval LNs**. Mistargeting of PN dendrites was analyzed based on a blind scoring method. Confocal stacks of brains from control and experimental groups were randomized by Researcher #1, analyzed by Researcher #2 and then decoded by Researcher #1.

**3D reconstruction of antennal lobes**. Control and corresponding DTI expression brains were shuffled by Researcher #1. Six brains of each group were blindly selected by Researcher #1 and subjected to Imaris (Bitplane) for 3D reconstruction. The surface of individual glomeruli, ALs and medial lobes of mushroom body was manually outlined based on neuropil marker, Brp by Researcher #2. Reconstructed brains were then re-examined by Researcher #3. The origin for X, Y, and Z-axes was set at the tip of the MB peduncle. The Y–Z plane was set along the brain midline, to evenly divide the two hemispheres, with Y-axis parallel to the MB dorsal lobe. Accordingly, Z-axis was perpendicular to the MB dorsal lobe. The X-axis was set perpendicular to Y–Z plane (Supplementary Fig. 11). The volumes of the AL and glomeruli were normalized to the volume of the MB medial lobe from the same brain hemisphere. The position and distances between glomeruli and AL were normalized to the length between the lateral edge of peduncle (reference

origin) and the center of medial lobe from the same brain hemisphere. Identities of reconstructed brains were then decoded by Researcher #1.

**Principle component analysis and individual variables**. Ten variables ($x$-position, $y$-position, $z$-position, distance to origin, ellipticity-oblate, ellipticity-prolate, sphericity, volume, area, mean of Brp intensity) of individual ALs were extracted from brains that had been 3D-reconstructed in Imaris. Except for ellipticity-oblate, ellipticity-prolate, and sphericity, the individual variables describing each AL were normalized to the MB from the same brain, as described above. For principle component analysis (PCA), the values of mean Brp intensity were further divided by 65535 such that all values fall between 0 and 1. The values for the rest of the nine variable groups were normalized to the maximal values within all analyzed brains, such that each value was not greater than 1. Normalized variables were then subjected to PCA, coded in the C++ language (Fig. 6g). The mean distances between any two glomeruli in a given genotype were calculated and represented in a heat-map matrix (Fig. 6h, i, Supplementary Fig. 12). The same ten variables ($x$-position, $y$-position, $z$-position, distance to origin, ellipticity-oblate, ellipticity-prolate, sphericity, volume, area, mean of Brp intensity) of individual glomeruli were extracted from all 3D-reconstructed ALs (Supplementary Table 4). To quantify the shift of individual glomeruli in $LN > DTI$ brains, the shift deviation between control and $LN > DTI$ was calculated for each glomerulus by the following formula: deviation = average $y$-position of DTI−average $y$-position of control (Fig. 6j, Supplementary Fig. 8e, g).

**Statistics**. Data shown in Figs. 3h, 6f, S7c, S7e, S7g, S8a and S8b were analyzed by two-tailed Student's $t$-test. Detailed values can be found in Supplementary Table 3 and 4.

**Script used in this study**. The script used in this study will be available upon request.

**Data availability**. The authors declare that all data supporting the findings of this study are available within the article and its Supplementary Information files or from the corresponding authors upon reasonable request.

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

## Acknowledgements

We thank Dr. Thomas Clandinin for sharing the InSITE GAL4 lines that were established in his lab. The GAL4 screen was initiated in Dr. Liqun Luo's lab by Y.H.C. Y.H.C. thanks Dr. Luo for his generosity to allow us continue this project. We thank Drs. Kristin Scott and Christoph Scheper for sharing the expression information for a subset of InSITE GAL4 lines, Shih-Yaw Yang for helping with the preliminarily characterizing developmental GAL4 expressions, Christopher Potter for sharing the *nSybQF2* fly before publication, Liqun Luo, Richard Benton, Julie Simpson, Henry Y. Sun, Chi-Kuang Yao, and Guang-Chao Chen for sharing fly strains and antibodies. We thank Drs. Liqun Luo, Marcus Calkins and Mr. Kai Hsiang Chang for feedback. We thank Kyoto, Bloomington, and NIG-FLY stock centers for fly stains, and Developmental Studies Hybridoma Bank for antibodies. This work was supported by a NSC undergraduate research fellowship (101-2815-C-001-024-B) to N.F.L, MOST grants (101-2311-B-001-016-MY3 and 104-2311-B-001-033-MY3) to Y.H.C, and a Career Development Award (AS-102-CDA-L02) to Y.H.C.

## Author Contributions

Y.H.C. designed experiments. T.Y.L., Y.J.C., Y.H.C., and S.H.L. performed the *GAL4* screen. N.F.L., Y.J.C., Y.H.C., S.H.L., T.H.W., C.J.Y., T.Y.L., and H.J.L. performed LN development experiments. S.H.L. performed experiments estimating the number of larval LNs and quantified LN numbers in adult brains. N.F.L. performed Kaede experiments. Y. H.C. and N.F.L performed EcR-related experiments. N.F.L., C.J.Y., and Y.H.C. characterized single LN morphologies. Y.H.C. scored LN innervation patterns and performed hierarchical clustering. Y.J.C. and N.F.L. examined the interaction between LNs and PNs. Y.J.C., C.J.Y., and Y.H.C. performed *670*-related experiments. Y.H.C, Y.T.C., T.H.W., H.J. L., and Y.J.C. performed PN dendrite or ORN axon targeting experiments. N.F.L., Y.H.C., and K.T.T. perform 3D reconstruction of ALs. K.T.T. designed, wrote the script for and performed PCA analysis and variables analysis. D.M.G. and M.A.S. provided InSITE GAL4 lines. Y.H.C. supervised the project and wrote the manuscript.

## Additional information

**Competing interests:** The authors declare no competing interests.

