## [Peer Review File · Nature Communications]

Reviewers' comments:

Reviewer #1 (Remarks to the Author):

In this study Dr. Chou and colleagues address an important developmental neuroscience question of how local interneurons are wired into neuronal circuits. Using a set of sophisticated genetic tools, they show that local interneurons in the adult antennal lobe – the first olfactory structure where sensory neurons synapse onto projection neurons that send sensory information to higher brain centers – can be separated into two major classes in *Drosophila*. During the transition from larval to adult stage, local interneurons either undergo apoptosis or are reintegrated into the adult neural circuit. Other local interneurons are born during metamorphosis. The data are of high quality and the interpretation is convincing. I am supporting the publication of the study. But I have a few minor concerns.

1. It is not clear whether male or female brains were used for the study. If both sexes were used, it would be important to know whether there is any sexual dimorphism.
2. Related to Figure 1. The text description seems to give the illusion that all 25 candidate lines mark GABAergic local interneurons. In fact, they are not. Some are GABAergic and others are not. For those negative for GABA immunostaining, their neurotransmitter identity is unknown. This needs to be made clear.
3. Related to Figure 5. It is not clear what the PCA shows. From what I can see, killing the 449-positive larval LNs increases the distance between glomeruli mostly in the anterior-posterior axis, whereas killing the 670-positive larval LNs has little or no effect on glomerular distance. Furthermore, labeling the X-, Y-, Z-axis in the figure would be more informative (at this moment, one has to go the methods section to figure that out). The glomerulus index is not found anywhere, which should be shown in a table in the supplemental data. It would be important to highlight the distance between neighboring glomeruli, because the distance between non-neighboring glomeruli reflects an additive effect, which is apparent in Figure 5l and m.
4. For all the experiments involved UAS-DTI or other UAS constructs, both the GAL4 and UAS lines should be used as controls. At the moment, only the GAL4 line has been used as control.
5. The stamen in the introduction “The *Drosophila* olfactory circuit shares similar organizational principles with those of mammals but it is numerically simpler” has just one reference. That is obviously an oversimplification.
6. Figure 1. The labels D (dorsal) and L (lateral) are missing. They are in the legend.

Reviewer #2 (Remarks to the Author):

This paper investigates the development of local interneurons (LNs) in the *Drosophila* antennal lobe (AL). The authors make three major claims. First, that LNs innervate the AL in three waves, comprising a subset born during larval stages, a subset that innervates the AL just prior to ORN-PN connectivity is established (~mid-pupal development), and a subset that innervates the AL after ORN-PN connectivity is established (late pupal development).

Second, they claim that specific types of larval LNs undergo pruning and regrowth into particular regions of the AL or cell death. And third, they claim that LN development shapes the global organization of the AL.

The novelty of the first claim, that different populations of LNs innervate the AL at different stages of development, seems limited given that previous research has shown that LNs are born throughout larval and pupal development. Significance is also limited as specific LN subtypes are not identified and characterized.

That some larval LNs reintegrate into the AL after pruning was not previously known. However, the authors did not identify and characterize specific LN subtypes that display this behavior. For example, an understanding of their mature morphologies and patterns of glomeruli innervation could provide clues as to their roles within the olfactory circuit and thus could considerably increase the significance of this finding.

The third claim, regarding the importance of LN development for AL organization is a bit vague and lacking specific controls. UAS constructs are leaky, and the disorganization observed may not be specific to ablation of the LNs in question. A UAS-DTI only control should be performed. Also, as this disorganization does not seem to affect connectivity, the significance of this observation is unclear.

I feel that the paper would be considerably strengthened by a more detailed description of LN subtypes (just focusing on the larval LNs would be good) and their specific patterns of innervation during development and in the mature circuit.

Reviewer #3 (Remarks to the Author):

In this paper, Lin et al perform a comprehensive study of LN development in the olfactory system in *Drosophila*. While LN are known to be important across animal kingdom, their development is relatively unstudied. Therefore, the paper focuses on an interesting question and reveals some interesting aspects of LN biology such as the fact that some classes undergo EcR dependent neuronal remodeling during metamorphosis while other presumably undergo apoptosis while still other classes are adult specific. All in all, I think the work is interesting and could be a good candidate for publication in *N. Comm.*

Specific comments:

The strong part of the paper are figure 1-3 where the developmental description is nice and interesting. A few potential improvements could be here:

- to directly demonstrate apoptosis as the mechanism of LN death, one could use anti cleaved caspase antibody (there are a couple working quite well).
- When discussing the Gal80ts section - as the half life of Gal80TS is substantial, I would write these sections a bit more carefully - something like: we were able to restrict the expression of EcRDN to early metamorphosis by shifting the temp to 29°C from mid-3L to 24h APF.

Figure 4 is interesting but inconclusive. I am not sure i understand why morphology is equated with identity, and would keep the more conservative morphology parameter here.

By far, the weaker part of the paper is the last figure. I am not sure I understand the logic of the choice of LN classes that are presumably killed. I also find it surprising that such a short temperature shift is sufficient to kill the LNs, but that said, the effect on glomerular location is rather modest. The analysis is confusing and should be deepened or at least explained better (annotation is really partial for this entire figure). In the very last part of the paper, the authors show us negative data but then still conclude that LN positioning helps maintain the structure of the AL. The reasoning for that claim is unclear. What is the bottom line here and why?

All in all, figures 1-3 form a strong core for a N. Comm paper. This could either compromise a short report, or - expanded to better understand the mechanisms of pruning/regrowth, or - could be presented in the present form but with modifications made to the presentation and textual conclusions from figures 4+5.

We thank all three reviewers and the editor for carefully evaluating our manuscript, for their appreciation of our work, and for their constructive criticisms. We have made substantial changes to our manuscript in response. Those changes are highlighted in blue in the revised manuscript. Below we provide a point-by-point response to the specific criticisms of the reviewers. We believe that we have satisfactorily addressed most of these criticisms. As a consequence, our paper is much stronger with increased depth.

Below, we organize our responses numerically and in blue for convenient cross-referencing. The original review is copied in full in black.

Reviewer #1 (Remarks to the Author):

In this study Dr. Chou and colleagues address an important developmental neuroscience question of how local interneurons are wired into neuronal circuits. Using a set of sophisticated genetic tools, they show that local interneurons in the adult antennal lobe – the first olfactory structure where sensory neurons synapse onto projection neurons that send sensory information to higher brain centers – can be separated into two major classes in *Drosophila*. During the transition from larval to adult stage, local interneurons either undergo apoptosis or are reintegrated into the adult neural circuit. Other local interneurons are born during metamorphosis. The data are of high quality and the interpretation is convincing. I am supporting the publication of the study. But I have a few minor concerns.

1. It is not clear whether male or female brains were used for the study. If both sexes were used, it would be important to know whether there is any sexual dimorphism.

#1. We thank the reviewer for his/her appreciation of our work. Indeed, it was previously demonstrated that a subset of Fru-positive interneurons undergoes differential developmental programs and cell death in male and female brains, leading to sexual dimorphic innervation patterns (Kimura et al. 2005. *Nature*. 438, 229-233). We fully agree that sexual dimorphism is an important mechanism that is likely to be exhibited in olfactory LNs. Unfortunately, except for the InSITE GAL4 screen that was performed exclusively on female brains, we did not separately analyze male and female brains in the rest of this study. We have included this information in the Methods (P. 13, line 6-7) and made a statement about sexual dimorphism in the Discussion (P. 12, line 16-21).

2. Related to Figure 1. The text description seems to give the illusion that all 25 candidate lines mark GABAergic local interneurons. In fact, they are not. Some are GABAergic and others are not. For those negative for GABA immunostaining, their neurotransmitter identity is unknown. This needs to be made clear.

#2. We thank the reviewer for pointing out this misleading statement. We have added a statement to make it clear: "Some of the identified GAL4 drivers label GABA-negative LNs (Supplementary Table 1), and the neurotransmitter identities of these neurons are currently unknown. " (P. 4, line 2-3).

3. Related to Figure 5. It is not clear what the PCA shows. From what I can see, killing the 449-positive larval LNs increases the distance between glomeruli mostly in the anterior-posterior axis, whereas killing the 670-positive larval LNs has little or no effect on glomerular distance. Furthermore, labeling the X-, Y-, Z-axis in the figure would be more informative (at this moment, one has to go the methods section to figure that out). The glomerulus index is not found anywhere, which should be shown in a table in the supplemental data. It would be important to highlight the distance between neighboring glomeruli, because the distance between non-neighboring glomeruli reflects an additive effect, which is apparent in Figure 5l and m.

#3. We apologize that the explanation of the PCA analysis results was too brief. We tried to condense our presentation of these results because of word count limitations and also the results are negative. However, we included the data because we thought it was useful to demonstrate the power of the positive results in subsequent analyses. After further consideration, we have deleted the PCA analysis of glomeruli but kept the PCA analysis of ALs (current Fig 6g). We also provide a more detailed statement of PCA analysis in the Methods (P. 17, line 1-7).

#4. After including *UAS-DTI* controls (also see our response #8) and re-analyzing all 3D-reconstructed ALs, we found that *449-GAL4* control brains are likely to exhibit a mild but significant abnormality in AL organization. We are therefore more conservative about interpreting the effect of killing 449-positive larval LNs (Supplementary Fig. 7a, 7b, and 8). The fact that ablating *NP3056*-positive larval LNs leads to glomerular shift still holds true. Accordingly, we moved the *NP3056* results to Fig. 6 and provide a more detailed analysis in Supplementary Fig. 7. The corresponding statements in the Results (P. 8 line 25- P. 9 line 11) and Discussion (P. 12 line 1-8) have been revised based on the new analyses. The conclusion that killing 670-positive larval LNs has little effect on AL organization still holds.

#5. We thank the reviewer for the insightful suggestion to include the X-, Y-, Z- axes labels in corresponding figures. We have done this in the new versions of Fig. 6J and Supplementary Fig. 7f, h.

#6. We agree that the information regarding the glomerular index will be very useful for readers to fully digest the analyses and dig out additional information. We have included such information as Supplementary Note 4 (Supplementary information P. 3).

#7. The reviewer suggested denoting the distances between neighboring glomeruli as opposed to non-neighboring glomeruli. The latter distances are likely to exhibit larger additive effects of small differences between neighbors. In principle it is a wonderful idea to do such a comparison. However, the glomerular index in Fig. 6h is arranged as groups of glomeruli (i.e., DA group, DL group, VA group, etc.). The glomerular index in Fig. 6j represents the distance between individual glomeruli and the reference origin. Neither of

these arrangements contains information about which two glomeruli may be neighbors. Furthermore, the irregular shapes of individual glomeruli make defining the neighbors of each glomerulus challenging. However, this suggestion inspired us to further analyze whether the changes in Fig. 6j correlated to the changes in Fig. 6h. The way we accomplished this goal was to re-align the data in Fig. 6h based on the glomerular index used in Fig. 6j. If these two types of changes exhibit some relationship (i.e., the changes between any two glomeruli are together reflected in the y-shift of two groups of glomeruli or vice versa), we should see clusters of "red" and clusters of "blue" in this analysis. Our results show no obvious clustering in such a matrix (Supplementary Fig. 11d), suggesting that killing NP3056 larval LNs might produce two independent effects on glomerular organization. We have described this new result in the Results (P. 9, line 4 - 7) and Discussion (P. 12, line 4-5).

4. For all the experiments involved UAS-DTI or other UAS constructs, both the GAL4 and UAS lines should be used as controls. At the moment, only the GAL4 line has been used as control.

#8. We thank reviewers 1 and 2 for making this same comment. We have conducted the *UAS-DTI* experiment. To implement blind comparisons in the 3D reconstruction of *UAS-DTI* control brains, one researcher shuffled the brains of each group (*GAL4* controls, *UAS-DTI* control and *LN>DTI*) and picked one brain from each of the previous 6 experimental groups and 6 brains from the new *UAS-DTI* group. The twelve brains were scrambled again and then passed to a second researcher to conduct 3D-reconstruction. The results were re-examined and extracted by researcher 3, decoded by researcher 1 and analyzed by researcher 1.

Regarding *UAS* controls for the rest of experiments, because *GAL4* drivers are required to visualize LNs, it is not feasible to conduct such controls for *LN>EcRDN* (Fig. 3d, Supplementary Fig. 3b), *670>p35* (Fig. 3g) or the number of larval LNs in *LN>DTI* (Supplementary Fig. 6).

5. The stamen in the introduction "The *Drosophila* olfactory circuit shares similar organizational principles with those of mammals but it is numerically simpler" has just one reference. That is obviously an oversimplification.

#9. We agree with the reviewer that the statement is over simplified. To offer better comparisons between *Drosophila* and mammal olfactory circuits by potential readers, we have included additional references and rephrased this statement as "The *Drosophila* olfactory circuit shares similar organizational principles with those of mammals in the first olfactory information processing center (i.e., olfactory bulb in mammals and antennal lobe in fly) but it is numerically simpler". (P. 2, line 23-25).

6. Figure 1. The labels D (dorsal) and L (lateral) are missing. They are in the legend.

#10. We thank the reviewer for noting this omission. We have corrected it.

Reviewer #2 (Remarks to the Author):

This paper investigates the development of local interneurons (LNs) in the *Drosophila* antennal lobe (AL). The authors make three major claims. First, that LNs innervate the AL in three waves, comprising a subset born during larval stages, a subset that innervates the AL just prior to ORN-PN connectivity is established (~mid-pupal development), and a subset that innervates the AL after ORN-PN connectivity is established (late pupal development). Second, they claim that specific types of larval LNs undergo pruning and regrowth into particular regions of the AL or cell death. And third, they claim that LN development shapes the global organization of the AL.

The novelty of the first claim, that different populations of LNs innervate the AL at different stages of development, seems limited given that previous research has shown that LNs are born throughout larval and pupal development. Significance is also limited as specific LN subtypes are not identified and characterized.

#11. We thank the reviewer for bringing up this point. Indeed, previous studies have demonstrated LNs are born from embryonic to late pupal stages. However, these experiments were conducted using MARCM (Chou et al. 2010. *Nat. Neurosci.* 13, 439-449), dual-expression-control MARCM (Lai et al. 2008. *Development* 135, 2883-2893) or twin-spot MARCM (Lin et al. 2012. *PLoS Biol.* 10, e1001425). Therefore the previous studies provided only the birth time of distinct types of LNs and no information about how and when those LNs develop and integrate to the circuit. Our work clearly bridges this gap. On top of this, we found a certain time delay between the birth of a given LN and the emergence and integration of the LN to the circuit. In addition, not counting embryonic born LNs, at least 48 lateral LNs are born during the larval stage and remain in adult brains (Lin et al., 2010. *PLoS Biol.* 10, e1001425). However about 22 larval LNs survive metamorphosis and remain in the AL of 24 h APF pupae. In other words, at least 26 adult specific LNs (after deducting 22 LNs from 48 LNs born at the larval stage plus embryonic born LNs) in the lateral cluster emerge and integrate to the circuit after 24 h APF. We have included this explanation as a Supplementary note 2.

#12. We thank the reviewer for his/her insightful criticism regarding the characterization of LN subtypes, which spurred us to examine how distinct subtypes of larval LNs re-integrate to the developing adult olfactory circuit. Because the single cell clone frequency is low and biased toward a subset of neurons, it is not feasible to study this issue through live imaging of pupal brains. We therefore dissected 315 *189Y* pupal brains and 434 *NP3056* pupal brains, covering four pupal developmental stages. We observed that distinct subtypes of larval LNs may progressively re-integrate into the developing adult antennal lobes throughout pupal development. We further did hierarchical clustering to show the heterogeneity of their innervation patterns in 72 h APF and adult brains. These data are shown in Fig. 5 and Supplementary Fig. 4 and described in the Results (P. 7 line 26-30, P. 8 line 6-8).

(Brief note: To concise the figure legends, we removed the description of larval LN cell types in previous Fig. 4 legend to the Supplementary Note 3.)

That some larval LNs reintegrate into the AL after pruning was not previously known. However, the authors did not identify and characterize specific LN subtypes that display this behavior. For example, an understanding of their mature morphologies and patterns of glomeruli innervation could provide clues as to their roles within the olfactory circuit and thus could considerably increase the significance of this finding.

#13. We thank the reviewer for clearly pointing out how we may increase the depth of this work. We have performed experiments to address this issue, as described in response #12. We hope this set of experiments will sufficiently increase the significance of our findings.

The third claim, regarding the importance of LN development for AL organization is a bit vague and lacking specific controls. UAS constructs are leaky, and the disorganization observed may not be specific to ablation of the LNs in question. A UAS-DTI only control should be performed. Also, as this disorganization does not seem to affect connectivity, the significance of this observation is unclear.

#14. Reviewer 1 also raised the same concern about *UAS-DTI* and other *UAS* controls. Please see our responses #8 and #4. Briefly, our new results demonstrate the effect of killing *NP3056* larval LNs is not extremely strong, but it is significant. Although we did not observe dendrite mistargeting in PNs, whether such glomerular disorganization would affect the connectivity among LNs is an open question. This is an interesting point of view, so we have included it in the Discussion (P.12, line 7-8).

I feel that the paper would be considerably strengthened by a more detailed description of LN subtypes (just focusing on the larval LNs would be good) and their specific patterns of innervation during development and in the mature circuit.

#15. We agree. Please see our responses #12 and #13.

Reviewer #3 (Remarks to the Author):

In this paper, Lin et al perform a comprehensive study of LN development in the olfactory system in *Drosophila*. While LN are known to be important across animal kingdom, their development is relatively unstudied. Therefore, the paper focuses on an interesting question and reveals some interesting aspects of LN biology such as the fact that some classes undergo EcR dependent neuronal remodeling during metamorphosis while other presumably undergo apoptosis while still other classes are adult specific. All in all, I think the work is interesting and could be a good candidate for publication in *N. Comm.*

Specific comments:

The strong part of the paper are figure 1-3 where the developmental description is nice and interesting. A few potential improvements could be here:

- to directly demonstrate apoptosis as the mechanism of LN death, once could use anti cleaved caspase antibody (there are a couple working quite well).
-When discussing the Gal80ts section - as the half life of Gal80TS is substantial, I would write these sections a bit more carefully - something like: we were able to restrict the expression of EcRDN to early metamorphosis by shifting the temp to 29°C from mid-3L to 24h APF.

#16. We thank the reviewer for expressing appreciation of our work and his/her comments on the first three figures. We have stained cleaved Caspase 3 in 24 h APF 670>DTI pupal brains. Indeed, we were able to observe some dying 670-positive LNs that actively express cleaved Caspase 3. The new result is shown in Fig. 3f, which is described in the Results (P. 6 line 23).

#17. Indeed, GAL80[ts] takes about three hours after heat-shock to allow GAL4 to drive reporter expression and it takes about 12 h after returning to a permissive temperature to fully suppress GAL4 activity (McGuire et al. 2003. Science 302, 1765-1768). We thank the reviewer for pointing out this issue and have re-phrased our statement in the Results (P. 5, line 33 - P.6, line 3) and Methods (P. 16, line 9-11).

Figure 4 is interesting but inconclusive. I am not sure i understand why morphology is equated with identity, and would keep the more conservative morphology parameter here.

#18. We agree with the reviewer that morphology may not equate to the identity of LNs; however, it is the most feasible and accessible parameter in this system to describe and distinguish *potentially* different subtypes of LNs. After following the suggestion of Reviewer 2 to examine how distinct subtypes of larval LNs integrate to the developing adult olfactory circuit (Fig. 5) (Responses #12 and #13), we hope the reviewer will agree that we have expanded our understanding of larval LN subtypes, which are often categorized by their morphologies.

By far, the weaker part of the paper is the last figure. I am not sure I understand the logic of the choice of LN classes that are presumably killed. I also find it surprising that such a short temperature shift is sufficient to kill the LNs, but that said, the effect on glomerular location is rather modest. The analysis is confusing and should be deepened or at least explained better (annotation is really partial for this entire figure). In the very last part of the paper, the authors show us negative data but then still conclude that LN positioning helps maintain the structure of the AL. The reasoning for that claim is unclear. What is the bottom line here and why?

#19. We apologize for not clearly presenting the glomerular organization experiments in our initial manuscript. We killed NP3056-positive larval LNs because they represent both

unilateral and bilateral LNs, and thus are probably more diverse than 189Y-positive larval LNs. In addition, The expression of *189Y-GAL4* in LNs is much weaker than that of *NP3056-GAL4* at early pupal stage. The reason we killed 449-positive larval LNs is because the population is large and presumably would show more severe phenotypes. Considering the half-life of GAL80[ts] and based on our analyses shown in Supplementary Fig. 6 b, d, we expect that DTI was expressed from wandering larval stage/early metamorphosis to about 12 h APF, which is likely to be sufficient to kill all three of the NP3056-positive larval LNs before they can re-extend neurites. Since only three LNs were killed, we believe we can attribute the weak but significant disorganization of glomeruli to the small number of killed LNs.

#20. Reviewer 1 raised the same comment about the annotation of glomerular index in Fig.6. We have corrected this (please see our response #6). The analyses shown in previous Fig. 5 (now Fig. 6) are indeed difficult to understand. We have used a simpler formula to quantify the deviation of y-shift (Fig 6 j, k, and Supplementary Fig. 7e, g, 8d) and explained this in the Methods (P. 17, line 12-15). We hope these modifications will make the analyses more understandable.

#21. In our initial manuscript, killing 670-positive larval LNs did not lead to significant y-axis shifts but produced mild changes in distance between some pairs of glomeruli (previous Fig 5k). After some consideration, we believe the subtitle we used is too strong and confusing. After including the new *UAS-DTI* controls and new analysis, we no longer see this mild effect (Fig. 6i, Supplementary Fig. 11). We have therefore changed the subtitle of the last part to be "Ablating dying larval LNs has minimal effects on AL geometry".

All in all, figures 1-3 form a strong core for a N. Comm paper. This could either compromise a short report, or - expanded to better understand the mechanisms of pruning/regrowth, or - could be presented in the present form but with modifications made to the presentation and textual conclusions from figures 4+5.

#22. We hope that by adding a new set of larval LN integration experiments (Fig. 5, Supplementary Fig 4), control experiments suggested by Reviewers, and improved analyses/annotation, the Reviewer will support publication of our manuscript as an article in *Nature Communications*.

REVIEWERS' COMMENTS:

Reviewer #1 (Remarks to the Author):

The authors have addressed my previous concerns and the revised manuscript is significantly improved. I continue to support the publication of this study.

Reviewer #2 (Remarks to the Author):

In the previous version of the manuscript the authors demonstrated that different types of larval LNs re-integrate into the adult circuit. My major suggestion to improve the manuscript was to go further and characterize the development of specific subtypes, and link this to the adult morphologies and glomerular innervation.

To address this the authors performed extensive characterization of single LNs using the 189Y and NP3056 drivers in fixed pupae at different stages of development. They demonstrate that specific re-extending LNs have somewhat specific innervation patterns from the outset, and that these differ between cell types. The limitation of this approach is that the same single LNs cannot be identified at each of the developmental stages, and so how specific LN types develop cannot be addressed. I appreciate the difficulty of live imaging, but the new analyses do not go much further to identify specific LN subtypes.

Reviewer #3 (Remarks to the Author):

In this revision the authors have addressed the vast majority of my comments as well as the other reviewers' points. I think the paper is significantly improved and in principle should be accepted. However, there are a few point remaining, that I think should be addressed - but I do not think they require further review by me.

1) The most crucial point is figure 6. I am really not sure what it is supposed to contribute to the paper. The AL is a neuropil structure comprised of axons and dendrites of three types of neurons - LN, PN and ORNs. If you take one subset of the LNs out - it is quite expected and not so interesting that the structure of the AL might slightly change. I don't think this means anything about the LNs contributing to global shaping of the AL. It just means that if you take out a group of cells that contribute to this structure - it's anatomy will likely change. It is up to the authors and editor to decide on this - but my take is scrap the figure.

2) The authors seemed to have corrected a previous error. In the first version of the manuscript they claimed to use the QF2-QUAS binary system but it seems that instead they used the hybrid LexAQF-LexAop system. I applaud the authors for finding and correcting this in time. However, the writing and some of the figures are now confusing. First, saying "A second method combining tubP-GAL80ts36 with two binary systems, GAL4-19 UAS and

QF-QUAS (or LexA:QF-LexAop2)" is incorrect. It seems they only used the hybrid (and yes, I do think it is important to mention this is a hybrid system - this was confusing to me) binary system which is called, based on the Potter lab who generated it the LexAQF-LexAop system. I believe the number 2 belongs to the QF, not the Aop but to be honest is not required here. Furthermore, the choice of this system is unclear - was there a reason? Regardless, and at the very least, the authors need to properly explain the system and fully spell its name. This is true also with the reporter - smGdP - which in reality is a non fluorescent reporter protein - tagged with V5 or HA in the case of this paper and also called Spaghetti Monster GFP. I believe this version is myristoylated - and thus membrane bound. Finally - especially as some of the figures make use of both mCD8GFP as well as the myr-smGdPV5/HA - I think it is super important to label correctly the figures as well as explain in details within the figure legends which reporter was used when - I think that 3e is erroneously labeled but it might not be the only one

3) in figure 3f - are the closeups from the same brain? I can't find the same number of cells - so if this is a subset of a stack this should be mentioned. If its not from the same brain then its weird.

We thank all three reviewers and the editor for carefully evaluating our manuscript. Below we provide a point-by-point response to the specific criticisms of the reviewers. We hope the reviewers will agree with our responses and support the acceptance of our paper for publication.

Major changes in the current revision:

1. Following the policy of Nature Communications to avoid "data not shown", we include the results of our experiments using *670*-driven *UAS-GAL4* and single cell clones of *449*-positive larval LNs in adult brains as Supplementary Figures 3c and 5, respectively.
2. To offer detailed information about the secondary antibodies used in this study, we include a new Supplementary Table 6.

In the point-by-point response, we leave the reviewers' comments in black and highlight our responses in blue.

Reviewer #1 (Remarks to the Author):

The authors have addressed my previous concerns and the revised manuscript is significantly improved. I continue to support the publication of this study.

#1. We greatly appreciate the reviewer for spending time to evaluate our work and for the continued support.

Reviewer #2 (Remarks to the Author):

In the previous version of the manuscript the authors demonstrated that different types of larval LNs re-integrate into the adult circuit. My major suggestion to improve the manuscript was to go further and characterize the development of specific subtypes, and link this to the adult morphologies and glomerular innervation.

To address this the authors performed extensive characterization of single LNs using the 189Y and NP3056 drivers in fixed pupae at different stages of development. They demonstrate that specific re-extending LNs have somewhat specific innervation patterns from the outset, and that these differ between cell types. The limitation of this approach is that the same single LNs cannot be identified at each of the developmental stages, and so how specific LN types develop cannot be addressed. I appreciate the difficulty of live imaging, but the new analyses do not go much further to identify specific LN subtypes.

#2. We thank the reviewer for appreciating our additional work, charactering single LN morphologies in distinct developmental pupal stages. Indeed, our current results do not allow us to unambiguously describe the development of LN subtypes through pupal stages. However, our new data show that some of the larval LNs have especially unique innervation patterns or extremely dense processes, such as the *NP3056*-positive bilateral

LNs (Supplementary Fig. 4a₁) and the second type of 189Y-positive LNs (Fig. 5a₂). We should at least recapture the neurite development of these subtypes of larval LNs.

We have added text to the Discussion, describing this limitation of our study and suggesting that live imaging of pupal brains may be applied to solve this issue in the future (P. 11, line 30-33, P. 12, line 1-2).

Reviewer #3 (Remarks to the Author):

In this revision the authors have addressed the vast majority of my comments as well as the other reviewers' points. I think the paper is significantly improved and in principle should be accepted. However, there are a few point remaining, that I think should be addressed - but I do not think they require further review by me.

#3. We thank the reviewer for his/her time and support of our paper.

1) The most crucial point is figure 6. I am really not sure what it is supposed to contribute to the paper. The AL is a neuropil structure comprised of axons and dendrites of three types of neurons - LN, PN and ORNs. If you take one subset of the LNs out - it is quite expected and not so interesting that the structure of the AL might slightly change. I don't think this means anything about the LNs contributing to global shaping of the AL. It just means that if you take out a group of cells that contribute to this structure - it's anatomy will likely change. It is up to the authors and editor to decide on this - but my take is scrap the figure.

#4. We thank the reviewer for this suggestion, but we prefer to leave Figure 6 as it is for three reasons.

(1) It has been shown that ablating a given class of ORNs or PNs at a late pupal stage does not significantly change glomerular organization. However, ablating both PNs and ORNs of the same class causes that particular glomerulus to exhibit a smaller size (Berdnik et al. 2006. *J. Neurosci.* 26, 3367-3376). Therefore, we were quite surprised to find that ablating 4 out of approximately 200 LNs at early pupal stage leads to mild but consistent changes in glomerular organization.

(2) Although *449-GAL4* alone has some effect on glomerular organization, killing 449-positive larval LNs leads to glomerular shifts that are different than those resulting from killing *NP3056*-positive larval LNs. Furthermore, premature ablation of 670-positive larval LNs does not cause glomerular organization defects.

(3) Currently, the mistargeting of PN dendrites or ORN axons is the most prevalent phenotypic criteria that is used to describe developmental defects in olfactory circuit wiring. We hope to offer an additional method to quantitatively describe defects of this circuit, particularly in the AL.

2) The authors seemed to have corrected a previous error. In the first version of the manuscript they claimed to use the QF2-QUAS binary system but it seems that instead they used the hybrid LexAQF-LexAop system. I applaud the authors for finding and correcting this in time. However, the writing and some of the figures are now confusing. First, saying

"A second method combining tubP-GAL80ts36 with two binary systems, GAL4-19 UAS and QF-QUAS (or LexA:QF-LexAop2)" is incorrect. It seems they only used the hybrid (and yes, I do think it is important to mention this is a hybrid system - this was confusing to me) binary system which is called, based on the Potter lab who generated it the LexAQF-LexAop system. I believe the number 2 belongs to the QF, not the Aop but to be honest is not required here. Furthermore, the choice of this system is unclear - was there a reason? Regardless, and at the very least, the authors need to properly explain the system and fully spell its name. This is true also with the reporter - smGdP - which in reality is a non fluorescent reporter protein - tagged with V5 or HA in the case of this paper and also called Spaghetti Monster GFP. I believe this version is myristoylated - and thus membrane bound. Finally - especially as some of the figures make use of both mCD8GFP as well as the myr-smGdPV5/HA - I think it is super important to label correctly the figures as well as explain in details within the figure legends which reporter was used when - I think that 3e is erroneously labeled but it might not be the only one.

#5. We thank the reviewer for reading our manuscript with careful attention to detail and are glad that he/she pointed out this confusion regarding QF2w and the LexAQF hybrid. We simultaneously worked on *QF2w-QUAS* and *LexAQF-lexAop2* systems when we received the flies. Since both systems worked very well in our hands, we presented the first result no matter whether it was from the *QF2w-QUAS* or *LexAQF-LexAop2* system. Briefly, experiments on 189Y-, NP3056- or 449-positive larval LNs are derived from the *LexAQF-LexAop2(-myr-smGdP-V5 or HA)* system (Fig. 3c, 4c, 4e, 5, Supplementary Fig. 4, 5). Experiments on 670-positive larval LNs were derived from *QF2w-QUAS(-mCD8GFP)* system (Fig. 3e). Experiments shown in Fig. 4b and 4d are direct flip-out clones, derived from *UAS>stop>myr::smGdp-HA (or -V5)*. So the labels shown in Fig. 3e and other figures (Fig. 3c, 4b-e, 5, and Supplementary Fig. 4, 5) are correct. The *LexAop2* flip-out flies used in this study were produced by Rubin's lab (Nern et al. 2015. PNAS. E2967-E2976) but not by Potter's lab so they are not "*LexAop*". Although the detailed genotypes can be found in Supplementary Table 5, the corresponding main text was indeed unclear and confusingly worded. Therefore, we have modified the text in several places (P. 5, Line 17-21, P. 6, Line 22-23, and P.16, Line 3-5). We have also corrected "smGdP" to be "myr::smGdp" in the text (P. 5, Line 23-26), schemes (Fig. 3b, 4a) and figure legends (Fig. 3, 4).

3) in figure 3f - are the closeups from the same brain? I can't find the same number of cells - so if this is a subset of a stack this should be mentioned. If its not from the same brain then its weird.

#6. We thank the reviewer for pointing out this issue. The bottom panels of Fig. 3f show a single confocal section from the same brain as of the projected stacks in the top panel. Due to the word limitation for figure legends, we did not describe this properly in the previous revision. We have now revised the legend and hope the editor will approve the extra words (p.26, line 3-4).